# The relationship between the gut microbiome and the risk of respiratory infections among newborns

Yuka Moroishi [1,2✉], Jiang Gui[2], Anne G. Hoen[1,2], Hilary G. Morrison[3], Emily R. Baker[4], Kari C. Nadeau [5], Hongzhe Li[6], Zhigang Li [7], Juliette C. Madan[1,8] & Margaret R. Karagas [1✉]

## Abstract

**Background** Emerging evidence points to a critical role of the developing gut microbiome in immune maturation and infant health; however, prospective studies are lacking.

**Methods** We examined the occurrence of infections and associated symptoms during the first year of life in relation to the infant gut microbiome at six weeks of age using bacterial 16S rRNA V4-V5 gene sequencing ($N = 465$) and shotgun metagenomics ($N = 185$). We used generalized estimating equations to assess the associations between longitudinal outcomes and 16S alpha diversity and metagenomics species.

**Results** Here we show higher infant gut microbiota alpha diversity was associated with an increased risk of infections or respiratory symptoms treated with a prescription medicine, and specifically upper respiratory tract infections. Among vaginally delivered infants, a higher alpha diversity was associated with an increased risk of all-cause wheezing treated with a prescription medicine and diarrhea involving a visit to a health care provider. Positive associations were specifically observed with *Veillonella* species among all deliveries and *Haemophilus influenzae* among cesarean-delivered infants.

**Conclusion** Our findings suggest that intestinal microbial diversity and the relative abundance of key taxa in early infancy may influence susceptibility to respiratory infection, wheezing, and diarrhea.

**Plain language summary**

The gut microbiome consists of all the microorganisms that live in the gut. It is known to have an impact on the maturation of the immune system; however, the impact of the infant gut microbiome on respiratory infections has not been fully investigated. This study investigated whether the gut microbiome present at an age of six weeks was associated with the number, and symptoms, of subsequent respiratory infections. We found that the number of different microorganisms and the amount of particular types in the gut were associated with the infants' risks of respiratory tract infections as well as symptoms such as wheezing. Our findings may contribute to developing interventions to modulate the microbiome and hence improve health.

[1] Department of Epidemiology, Geisel School of Medicine at Dartmouth, Hanover, NH, USA. [2] Department of Biomedical Data Science, Geisel School of Medicine at Dartmouth, Hanover, NH, USA. [3] Josephine Bay Paul Center for Comparative Molecular Biology and Evolution, Marine Biological Laboratory, Woods Hole, MA, USA. [4] Department of Obstetrics and Gynecology, Dartmouth-Hitchcock Medical Center, Lebanon, NH, USA. [5] Sean N. Parker Center for Allergy and Asthma Research, Stanford University School of Medicine, Stanford, CA, USA. [6] Department of Biostatistics, Epidemiology and Informatics, Perelman School of Medicine, University of Pennsylvania, Philadelphia, PA, USA. [7] Department of Biostatistics, University of Florida, Gainesville, FL, USA. [8] Department of Pediatrics, Children's Hospital at Dartmouth, Lebanon, NH, USA. ✉email: Yuka.Moroishi.GR@dartmouth.edu; Margaret.R.Karagas@dartmouth.edu

nfections remain the leading causes of mortality in infants globally[1]. The human gut microbiome is becoming increasingly recognized for its critical role in immune function and the inflammatory response[2,3]. A bidirectional relationship emerges following birth whereby the gut microbiome aids the maturation of the immune system and the immune system regulates host–microbe symbiosis[4,5]. The impacts of perturbing these intricate relationships are evident in high-risk infants. For example, among infants with cystic fibrosis, the composition of the gut microbiome is a determinant of colonization with opportunistic pathobionts[6,7]. Likewise, in preterm infants, the gut microbiome is associated with fatal occurrences of necrotizing enterocolitis and infection[2,8,9]. Factors driving the establishment of the gut microbiome, including delivery mode and breast feeding[3,10–13], have also been related to the risk of infections[3,14–18]. Furthermore, the use of antibiotics during pregnancy, which has been found to influence the gut microbiome of offspring[19–21], increased the risk of infant infection-related hospitalizations[22]. Encouraging results from probiotic trials suggest health benefits from altering the gut microbiome, including an enhanced immune response to pathogens[23,24]. While studies have found possible links between early gut microbiome composition and infant infection[3,25], few prospective studies have been conducted, particularly in the general population.

We report on gut microbiome diversity and composition among infants during the critical period of early immune training and the subsequent occurrence of respiratory infections and symptoms, such as wheezing and diarrhea, in the first year of life as part of a prospective study of a cohort of pregnant women and their offspring from the general population in New Hampshire. Here, we measure the fecal microbiome to measure the gut microbiome. Wheeze and diarrhea outcomes for this study include those of any cause. Based on ASV data generated from 16S rRNA sequencing, higher alpha diversity at 6 weeks of age is associated with having an additional respiratory infection or symptom of respiratory infection requiring a prescription medicine, with associations varying by delivery mode. Using next-generation sequencing (NGS), shotgun metagenomics, *Veillonella* in all deliveries and *Haemophilus* in cesarean deliveries are among the species in 6-week stool identified as being related to an additional subsequent respiratory infection or symptom of respiratory infection requiring a prescription medicine during an infant's first year of life.

## Methods

**Study population**. Participants included mother–infant dyads from the New Hampshire Birth Cohort Study from whom we obtained infant stool samples at approximately 6 weeks of age. Pregnant women aged 18–45 were recruited from prenatal clinics in New Hampshire, USA, starting in January 2009, as described previously[26]. Women who were living in the same household served by a private water system since their last menstrual period, had no plans to move, and had a singleton pregnancy were included in the cohort. Participants completed surveys on infant lifestyle questions such as feeding mode, solid food introduction, and daycare. Infant birth characteristics were ascertained from newborn medical records, and maternal characteristics were abstracted from prenatal and delivery records, including age at enrollment, prenatal use of antibiotics, and prepregnancy body mass index in kilograms per meter squared. The Committee for the Protection of Human Subjects at Dartmouth College approved all protocols, and participants provided written informed consent upon enrollment.

**Ascertainment of infant health outcomes**. Telephone interviews were conducted with infants' caregivers in the first year of life, i.e., when infants turned approximately 4, 8, and 12 months of age.

Caregivers were asked whether their child had any upper respiratory tract infections (RTIs) or associated symptoms (e.g., runny nose, stuffy nose, eye infection, ear infection, influenza, sinus infection, pharyngitis, or laryngitis), lower RTIs (e.g., bronchitis, pneumonia, bronchiolitis (including respiratory syncytial virus (RSV)), or whooping cough), acute respiratory symptoms (e.g., difficulty breathing, wheezing, fever, or cough), or diarrhea since the previous interview. For each positive response, participants were then asked whether the condition lasted more than 2 days, whether the child saw a physician, and whether the child received any prescription medications for the condition.

**Stool sample collection, DNA extraction, sequencing, and profiling**. We measured the fecal microbiome of infant stool as a measure of the infant gut microbiome. Infant stool samples were collected at regularly scheduled ~6-week postpartum follow-up appointments as described previously[13,27]. Samples were aliquoted and frozen at −80 °C within 24 h of receipt. A Zymo DNA extraction kit (Zymo Research) was used for DNA extraction from thawed samples, and an OD260/280 nanodrop was used to measure sample quality and purity. The V4-V5 hypervariable region of the bacterial 16S rRNA gene was sequenced using Illumina MiSeq at the Marine Biological Laboratory in Woods Hole, MA. Amplicon sequence variants (ASVs) were then inferred using DADA2[28], and taxonomies were assigned using the SILVA database[29]. Quality control measures were conducted as described previously[13]. A subset of stool samples was also sequenced with NGS and shotgun metagenomics sequencing as previously described[21]. Extracted DNA samples were sheared to a mean insert size of 400 bp using a Covaris S220 focused ultrasonicator. The sequencing libraries were constructed using Nugen's Ovation Ultralow V2 protocol, and samples were sequenced using Illumina NextSeq. DNA reads were merged and trimmed using KneadData[30] for quality control before species-level taxonomic profiles were generated using Metaphlan2[31].

**Statistics and reproducibility**. We examined the association between gut microbiome composition and health outcomes ascertained during interval interviews over the subjects' first year of life. For our analyses, we examined the total number of reported outcomes, specifically upper RTIs and lower RTIs, as well as symptoms such as wheezing with a reported visit to a physician and treatment with a prescription medication. Diarrhea is not typically treated with prescription medications in infants; therefore, we focused the analyses on reports of diarrhea that involved a doctor visit. We imputed missing outcomes if the caregiver completed the interview but a specific question was unanswered using multiple imputation by chained equations and the predictive mean matching method.

For models using 16S data, we aggregated ASVs to the genus level and calculated alpha diversity on read counts per genus using the inverse Simpson index. We then used generalized estimating equation (GEE) for repeated measures with Poisson regression and AR(1) correlation structure to assess the association between $\log_2$-transformed 16S-based genus-level alpha diversity and each of the outcomes of interest. For models using metagenomics species data, we calculated the $\log_2$-transformed relative abundance using a pseudocount of $5 \times 10^{-20}$ for zero values. We also used a GEE for repeated measures with Poisson regression and AR(1) correlation structure to estimate relationships with species present in at least 10% of subjects. We applied a false discovery rate (FDR) threshold of 0.1 to adjust for multiple testing[32]. Factors associated with both the gut microbiome and health outcomes were considered potential

confounders and included in all GEE analyses. These confounders included maternal prepregnancy body mass index (BMI) (kg/m$^2$), delivery mode (vaginal/cesarean), infant sex (male/female), breast feeding at six weeks (exclusively breastfed/mixed fed or exclusively formula fed), antibiotic use during pregnancy (yes/no), and gestational age (complete weeks). We also conducted stratified analyses by delivery type for both alpha diversity using 16S data and microbial species based on metagenomics data. Due to sample size limitations, we conducted stratified analyses only for species-specific analyses on all outcomes combined and for upper RTIs.

For interpretability, we exponentiated the coefficient values to obtain relative risk (RR) and 95% confidence intervals (CIs). Of the 465 participants included in the 16S analyses, 391 participants (84.1%) had complete data for all potentially confounding variables. Of the 185 participants included in the metagenomics analyses, 160 participants (86.5%) had complete data for all potentially confounding variables. We assumed missing confounder entries were missing at random and used multiple imputation by chained equations and the predictive mean matching method to impute missing data. All analyses were performed using R version 3.4.3 using the functions *diversity*, *mice*, and *geeglm* in the 'vegan', 'mice', and 'geepack' packages.

**Reporting summary**. Further information on research design is available in the Nature Research Reporting Summary linked to this article.

## Results

**Baseline characteristics**. As of July 2019, we had completed 16S rRNA V4–V5 hypervariable region gene sequencing on 513 infant stool samples and whole-genome metagenomics sequencing on 202 infant stool samples collected at approximately 6 weeks of age. After removing infants for whom health information was unavailable in telephone surveys, our analysis included 465 infants with 16S data and 185 infants with metagenomics data. Our study population had an approximately equal distribution of male (53.4%) and female infants (46.6%) (Table 1). Nearly half of the infants (56.2%) had been exclusively breastfed at approximately 6 weeks of age, and approximately one-fifth of mothers (18.5%) had reported antibiotic use during pregnancy (Table 1). Cesarean section deliveries accounted for one-third of deliveries (30.3%) (Table 1). The five most common genera in our 16S data were *Escherichia/Shigella*, *Bacteroides*, *Bifidobacterium*, *Klebsiella*, and *Enterococcus* (Supplementary Fig. 1). The five most common species in our metagenomics data were *Bifidobacterium longum*, unclassified *Escherichia* species, *Escherichia coli*, *Bifidobacterium breve*, and *Gemella haemolysans* (Supplementary Fig. 1). The numbers of each respiratory infection and symptom at each age are provided in Supplementary Tables 1 and 2.

**16S V4–V5 rRNA gene: alpha diversity**. Associations were determined via the Wald test in GEE analyses with a *p*-value threshold of 0.05. Alpha diversity was positively associated with the occurrence of any respiratory infection or symptom of respiratory infection, which included upper RTIs, lower RTIs, and acute respiratory symptoms. Upper RTI outcomes were specifically associated. Each doubling in alpha diversity was associated with a 39% increase in having an additional respiratory infection or symptom of respiratory infection (RR = 1.39, 95% CI: 1.1–1.77) and a 40% increase in an additional upper RTI (RR = 1.40, 95% CI: 1.12–1.76) (Fig. 1, Supplementary Table 3). Among vaginally delivered infants, a doubling of alpha diversity was

associated with a 62% increase in having an additional respiratory infection or symptom of respiratory infection (RR = 1.62, 95% CI: 1.23–2.15) (Fig. 1, Supplementary Table 3). A doubling of alpha diversity was associated with a doubling of the risk of wheezing for which a medication was prescribed (RR = 2.00, 95% CI: 1.16–3.45) and an 86% increase in diarrhea requiring a doctor visit (RR = 1.86, 95% CI: 1.14–3.03) among vaginally delivered infants (Fig. 1, Supplementary Table 3). We did not observe consistent associations among cesarean-delivered infants.

**Metagenomics: species-level**. In the GEE of metagenomics species data, the doubling of the relative abundance of *Veillonella* unclassified was positively associated with having an additional respiratory infection or symptom of respiratory infection in the first year of life (RR = 1.02; 95% CI: 1.01–1.04) (Fig. 2a). In examining specific outcomes, we found that diarrhea was positively associated with the relative abundance of *Streptococcus peroris* and negatively associated with the relative abundance of *Streptococcus salivarius* (Fig. 3).

Stratified by delivery mode, we found that having an additional respiratory infection or symptom of respiratory infection was positively associated with *Haemophilus influenzae* among cesarean-delivered infants (RR = 1.02; 95% CI: 1.01–1.04) (Fig. 2c). *Veillonella parvula*, *Corynebacterium pseudodiphtheriticum*, and *Corynebacterium pseudodiphtheriticum* were positively associated, while *Clostridium butyricum* and *Coprobacillus unclassified* were negatively associated, with a risk of an additional upper RTI among infants delivered by cesarean section (Fig. 3).

## Discussion

In our prospective study of infants from the general population in New Hampshire, USA, we observed patterns of the early microbiome that were related to the occurrence of infant respiratory infections, wheezing, and diarrhea. Higher diversity of the early infant gut microbiome was associated with a greater number of respiratory infections and symptoms over the first year of life. Relationships between early microbial patterns and infant outcomes differed by delivery mode, a known contributor to the developing microbiome[13], with stronger associations with alpha diversity among infants born by cesarean section. Using metagenomic sequencing, we found that *Veillonella* in any delivery mode and *Haemophilus* in cesarean deliveries were among the species associated with an increased risk of infant respiratory infections and symptoms.

Our analyses found associations with many bacterial species that are commonly found in oral flora, although these bacteria have also been detected in the gut. An early driver of the gut microbiome is diet. One prospective study found that exclusive breastfeeding was inversely related to lower respiratory tract infections among infants and asthma and allergic rhinitis among children 4 years of age[33]. The same study highlighted the potential mediating effect of the gut microbiome on the relationship between exclusive breastfeeding and outcomes. Additionally, infants born operatively may have to acquire such species through breast milk[27].

In our study, *Veillonella*, specifically *Veillonella parvula*, was positively associated with upper respiratory infections, especially in cesarean-delivered infants. *Veillonella parvula* is commonly found in oral flora, although it is observed in both oral and gut ecosystems[34]. We are not aware of studies that have examined *Veillonella parvula*. However, consistent with our findings, a prospective study of 120 Dutch infants found an abundance of three *Veillonella* operational taxonomic units using 16S V4 rRNA sequencing among 1-week-old infants that was associated with a

**Table 1 Selected baseline characteristics of mothers and infants in the New Hampshire Birth Cohort Study.**

| Variable | 16S V4–V5 rRNA | | Metagenomics | |
|---|---|---|---|---|
| | Sample size | Mean (SD) or no. (%) | Sample size | Mean (SD) or no. (%) |
| *Maternal characteristics* | | | | |
| Age at enrollment, mean (SD), years | 465 | 31.9 (4.6) | 185 | 31.9 (4.3) |
| Body Mass Index before pregnancy, mean (SD), kg/m$^2$ | 462 | 25.8 (5.9) | 185 | 25.7 (5.7) |
| Parity, No. (%) | 461 | | 184 | |
| 0 | | 217 (47.1) | | 92 (50.0) |
| 1 | | 166 (36.0) | | 60 (32.6) |
| 2+ | | 78 (16.9) | | 32 (17.4) |
| Antibiotic use during pregnancy, No. (%) | 427 | | 173 | |
| Yes | | 79 (18.5) | | 36 (20.8) |
| No | | 348 (81.5) | | 137 (79.2) |
| *Infant characteristics* | | | | |
| Delivery mode, No. (%) | 465 | | 185 | |
| Vaginal | | 339 (72.9) | | 129 (69.7) |
| Cesarean | | 126 (27.1) | | 56 (30.3) |
| Infant sex, No. (%) | 464 | | 185 | |
| Male | | 248 (53.4) | | 107 (57.8) |
| Female | | 216 (46.6) | | 78 (42.2) |
| Breast feeding at 6 weeks, No. (%) | 427 | | 171 | |
| Exclusively breast fed | | 240 (56.2) | | 92 (53.8) |
| Mixed fed or exclusive formula fed | | 187(43.7) | | 79 (46.2) |
| Gestational age, Mean (SD), weeks | 465 | 39.1 (1.6) | 185 | 39.0 (1.7) |

Of the 465 mothers included in the 16S analyses, maternal BMI was missing for 3 mothers. Parity was missing for 4 mothers, and antibiotic use during pregnancy was missing for 38 mothers. Of the 465 infants included in the 16S analyses, infant sex was missing for 1 infant, and feeding type was missing for 38 infants. Of the 185 mothers included in the metagenomics analyses, parity was missing for 1 mother, and antibiotic use during pregnancy was missing for 12 mothers. Of the 185 infants included in the metagenomics analyses, feeding type was missing for 14 infants.
*SD* standard deviation, *No.* frequency, *kg* kilograms, *m* meters.

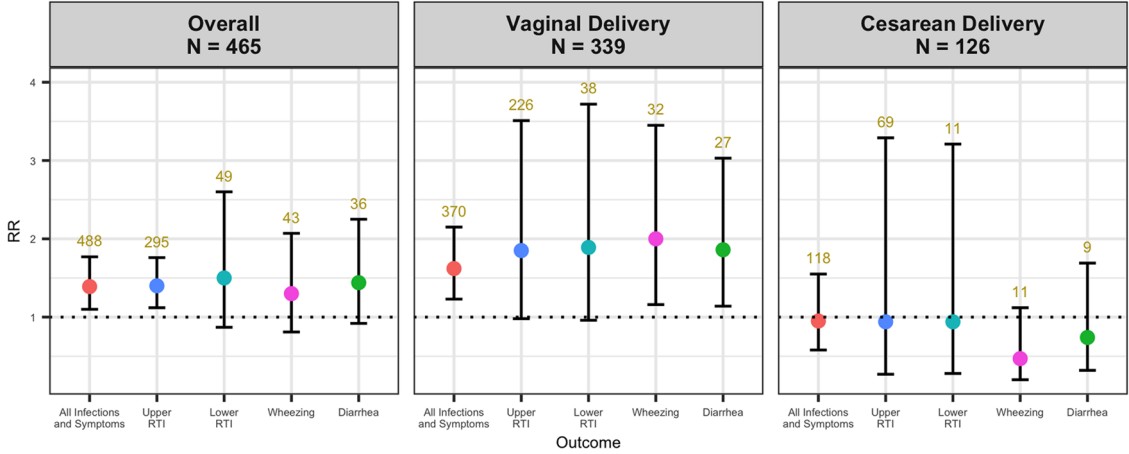

**Fig. 1 Dot and Whisker plots of adjusted relative risk estimates and 95% confidence intervals from GEE analysis of 6-week-old infant stool 16S V4–V5 rRNA sequencing alpha diversity and infections and symptoms of infection over the first year of life.** Overall GEE adjusted for maternal BMI, delivery type, sex, breast feeding at six weeks, perinatal antibiotic use, and gestational age. GEE stratified by delivery mode (vaginal and cesarean) adjusted for maternal BMI, sex, breast feeding at six weeks, perinatal antibiotic use, and gestational age. Red, blue, turquoise, pink, and green points represent relative risk for all infections and symptoms, upper RTI, lower RTI, wheezing, and diarrhea outcomes respectively. Vertical lines above and below points represent upper and lower confidence bands. Relative risk estimates represent an increased risk of having an additional infection or symptom of infection or an increased risk of experiencing wheezing or diarrhea with each doubling of the inverse Simpson index. Upper RTI, lower RTI, and wheezing outcomes are those diagnosed by a physician for which a medication was prescribed. Diarrhea outcomes are those diagnosed by a physician for whom no medication was prescribed. Numbers above upper confidence bands indicate the total number of outcomes, which may be greater than N due to repeated measures. Sample sizes were N = 464 for overall and N = 125 for cesarean delivery for diarrhea analyses due to missing data. GEE generalized estimating equation, *N* sample size, *RR* relative risk, *RTI* respiratory tract infection.

higher number of respiratory infections in the first year of life[3]. In mechanistic studies, *Veillonella parvula* produces propionate in the human gut, which may stimulate IL10-producing Treg differentiation[35,36], and in the small intestine, it induces IL-8, IL-1β, IL-10, and TNF-α[37].

Among cesarean-delivered infants in our study, a higher relative abundance of *Corynebacterium* species was associated with a greater risk of upper RTIs. *Corynebacterium* species are generally characterized as pathobionts in the respiratory tract[38]. Case series have suggested that *Corynebacterium pseudodiphtheriticum* in

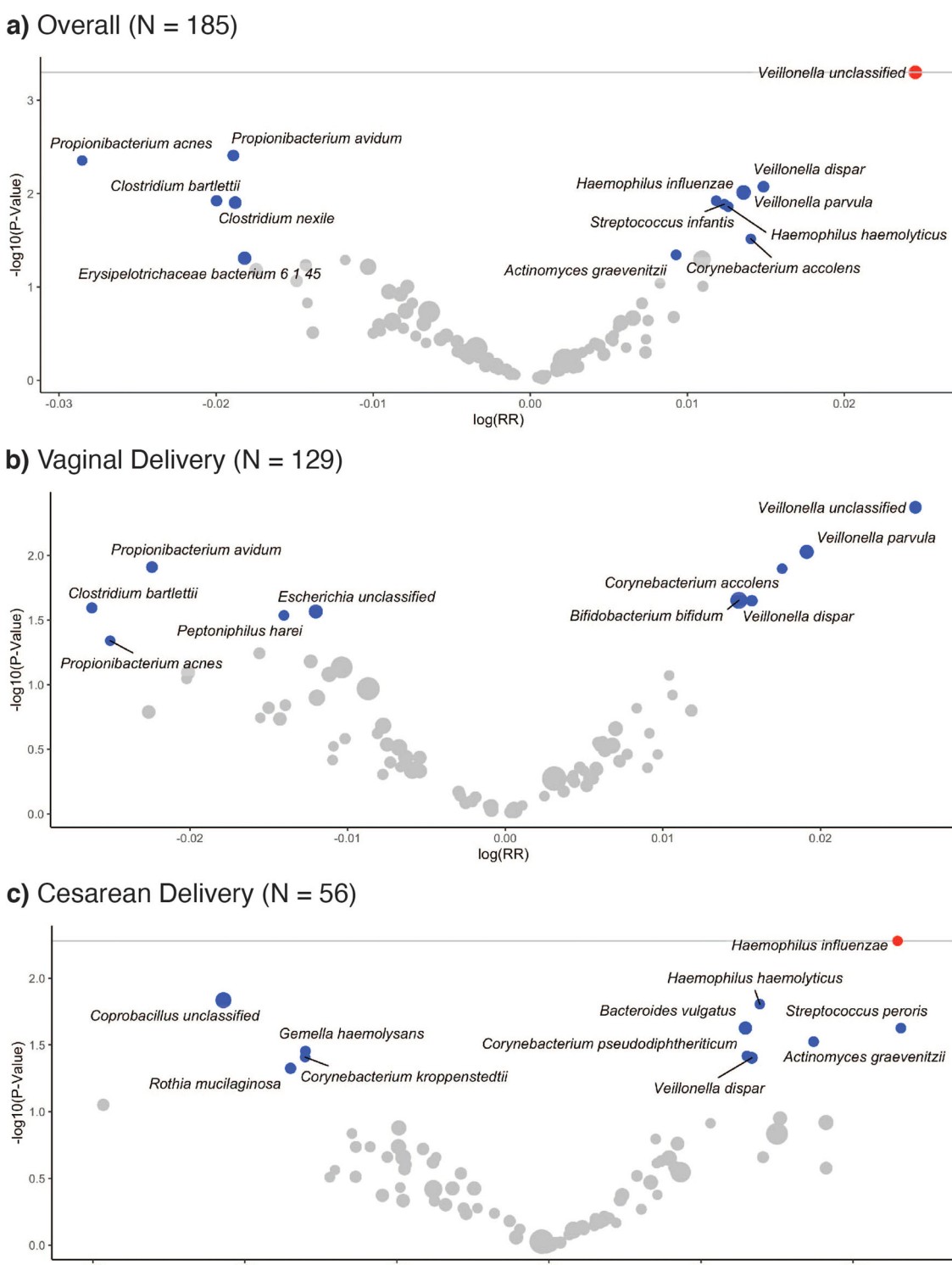

**Fig. 2 Volcano plots of GEE adjusted relative risk estimates of the number of infections and symptoms over the first year of life in relation to 6-week metagenomics species relative abundance.** Estimates shown for taxa prevalent in over 10% of subjects. The gray line represents a log10-transformed FDR threshold of 0.1. Blue points indicate statistically significant taxa at $\alpha = 0.05$. Red points indicate taxa selected by FDR correction. Gray points indicate all other taxa. The size of the points is scaled by relative abundance. Relative risk estimates represent an increased risk of having additional infections or symptoms of infection with each doubling of relative abundance. **a** Volcano plot of unstratified GEE adjusted for maternal BMI, delivery type, sex, breast feeding at six weeks, perinatal antibiotic use, and gestational age. **b** Volcano plot of vaginal deliveries adjusted for maternal BMI, sex, breast feeding at six weeks, perinatal antibiotic use, and gestational age. **c** Volcano plot of cesarean deliveries adjusted for maternal BMI, sex, breast feeding at six weeks, perinatal antibiotic use, and gestational age. Three taxa were removed due to high RRs and low *p*-values. GEE generalized estimating equation, *N* sample size, RR relative risk.

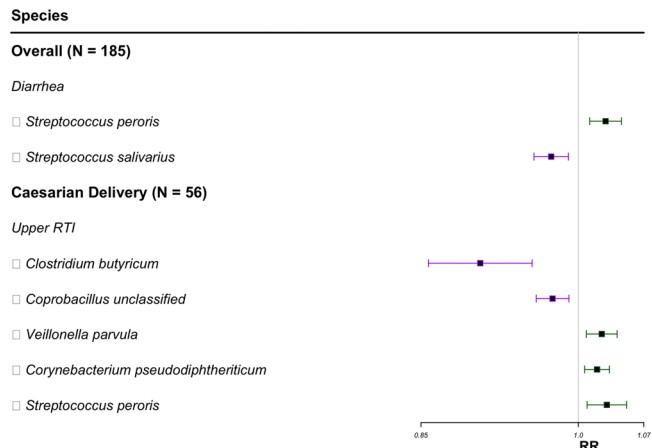

**Fig. 3 Forest plot of metagenomics species associated with the number of infections and symptoms of infections in the first year of life.** Species selected by FDR correction presented in the forest plot. Species for vaginal deliveries did not meet the FDR threshold of 0.1. Squares represent RR, and horizontal lines represent 95% confidence intervals. Green represents a positive association, and purple represents a negative association. Relative risk estimates represent an increased risk of having an additional upper respiratory infection or an increased risk of experiencing diarrhea with each doubling of the relative abundance. N sample size, RTI respiratory tract infection, RR relative risk.

sputum is a driver of pulmonary infection[39,40], and a case-control study of the nasopharyngeal microbiome from France found enriched *Corynebacterium pseudodiphtheriticum* in subjects with viral respiratory tract infections compared to healthy controls (N = 178)[41].

Upper respiratory tract infections in our cohort were associated with a decreased abundance of *Clostridium butyricum* in infant stool samples. One *Clostridium butyricum* strain upregulated inhibitory cytokines such as IL-10 in a mouse model[42]. Clostridium species can promote Treg production and inhibit inflammatory cytokines[43,44], with some associated with systemic infections in humans[45]. Thus, the potential inhibitory impact of *Clostridium butyricum* on an infant's immune response to infection requires further investigation.

Cesarean-delivered infants in our study had a positive association between *Haemophilus influenzae* and the number of any respiratory infections and symptoms. *Haemophilus influenzae* is a bacterial species known to cause several types of infectious diseases, including those of the respiratory tract. Although previous studies have not found associations between *Haemophilus influenza* in the gut microbiome and respiratory infections, the species do reside in the intestinal tract[46]. Further explorations of the gut-lung axis, as well as the origin of such bacteria in the gut, are warranted[47].

Other studies have also found associations between the gut microbiome and respiratory infections. Observations from epidemiologic studies that were not found in our study included a higher abundance of *Bifidobacterium* and *Enterococcus* and a lower abundance of *Escherichia-Shigella*, *Prevotella*, *Faecalibacterium*, and *Enterobacter* in subjects aged 0–3 years with pneumonia compared to healthy controls (N = 33) in a cross-sectional study of Mongolian children[48]. A case-control study of US infants found that infants with a higher gut alpha diversity of *Bacteroides* had a higher likelihood of developing bronchiolitis (N = 155)[49]. Findings from the aforementioned Dutch prospective study found several associations between the bacterial taxa of the infant gut microbiome and the number of respiratory infections, including associations with *Bifidobacterium*, *Bacteroides*, and

*Enterococcus* (N = 120)[3]. These findings, as well as ours, require further confirmation.

We found a positive prospective association between the overall alpha diversity at 6 weeks of life and the risk of upper RTIs. Although our findings for alpha diversity may seem contradictory to some prior studies that reported negative associations between alpha diversity and health outcomes[50], our study focuses on the early microbiome when diversity is low in healthy babies. We further prospectively examined associations with respiratory infections. Similar to our study, other studies found no associations between alpha diversity and adverse health outcomes but observed differences in the abundance of specific microbes[24]. A cross-sectional study from the US found that infants hospitalized for severe RSV infection had slightly lower alpha diversity of the gut than infants with moderate RSV infection and controls (N = 95)[25]. However, as the infants' gut microbiomes were assessed after the onset of disease, reverse causality was possible in this study; thus, further prospective studies are needed to understand how the overall microbial diversity and colonization of the neonatal and early infant gut reflect immune response to infections.

Wheezing is a respiratory symptom associated with infection, atopy, allergy or a later diagnosis of asthma. We found an association between a higher alpha diversity and an increased risk of wheezing identified by a physician, and this was largely among vaginally delivered infants. Epidemiologic studies have reported associations between the infant gut microbiome and atopic wheezing and asthma in childhood[51–54]. Whether our findings translate to a later risk of asthma will require longer-term follow-up of our cohort.

Among infants in our study, higher alpha diversity was associated with an increased risk of diarrhea requiring a physician visit in the first year of life. Other prospective studies are lacking, and to our knowledge, prior work includes only case-control studies that measured the infants' stool microbiomes at the time of symptoms. For example, in a small case-control study from South Africa, lower alpha diversity was observed in the stool of infants with gastrointestinal disease compared to infants with respiratory disease and infants with other diseases (N = 34)[55]. In addition to the issue of reverse causality, diarrheal diseases in the US vary in etiology and consequences compared to those in other geographic regions.

We observed that an increased relative abundance of *Streptococcus peroris* and a reduced relative abundance of *Streptococcus salivarius* were associated with a higher risk of diarrhea seen by a physician. Limited data also exist on microbiome composition in relation to diarrheal disease, again with most studies being cross-sectional and with relatively small sample sizes. A study from China of 20 infants with diarrheal illness and 13 controls found differences in gut microbiome composition, with two patients having a higher abundance of *Streptococcus peroris* than controls[56]. Other studies designed to detect pathogens among ill infants and young children compared to controls using 16S rRNA sequencing have also noted a higher abundance of *Streptococcus* species associated with diarrheal illness[57,58]. In mice, *Streptococcus salivarius* strains inhibited inflammation with severe and moderate colitis[59]. In the same experiment, *Streptococcus salivarius* inhibited the activation of the NF-κB pathway, which induces proinflammatory gene expression in intestinal epithelial cells[59]. Together, these findings raise the possibility of a role of *Streptococcus* species in susceptibility to diarrhea in early childhood.

Our study had a number of strengths as well as limitations. We carefully collected infant stool samples at approximately 6 weeks of age from a cohort of pregnant women and their offspring from the general population, and we examined repeated measurements

of infection occurrences, respiratory symptoms, and diarrhea and a broad range of potential confounding variables, such as maternal prepregnancy BMI, delivery mode, infant sex, breast feeding at 6 weeks, antibiotic use during pregnancy, and gestational age, for the analyses. A major challenge to the analysis of microbiome sequencing data is the ability to fully capture their correlated, compositional, and high-dimensional nature when assessing longitudinal outcomes. Therefore, we performed our analyses on the relative abundance of individual species and corrected for multiple hypotheses using the FDR. Our outcomes were not ascertained by viral or bacterial culture or PCR to confirm the type of infection; as a result, we relied on responses of telephone surveys from caregivers. Additionally, we could not differentiate between the various causes of wheezing and diarrhea in our dataset. Participant recall is a potential source of bias; however, efforts were made to reduce misclassification by including questions on the duration and severity of illness and limiting our analyses to outcomes that involved either a physician visit or a prescription medication. In a review of infants' pediatric medical records, we found caregiver responses to be at least 80% concordant with physician assessments documented in the medical record (unpublished data). Furthermore, while our study is one of the largest overall, we had limited statistical power in our analyses stratified by delivery mode.

In conclusion, our findings from a prospective birth cohort of US infants suggest that the composition of the microbiome in early life influences the most common health outcomes of infancy, which in turn may have consequences for lifelong disease risk. While higher alpha diversity was associated with respiratory infections and symptoms overall and among vaginal deliveries, the doubling of the relative abundance of unclassified *Veillonella* species and *Haemophilus influenza* species increased the risk of an additional respiratory infection or symptom overall and among cesarean-born infants, respectively. Our findings may help to inform interventions aimed at altering the microbiome during this critical window of immune training[60].

## Data availability

The microbiome data used in this study is publicly available in the Sequence Read Archive at http://www.ncbi.nlm.nih.gov/sra under the accession number PRJNA296814. The full study data is not publicly available due to their sensitive and identifiable nature; it may be made available upon request to the corresponding author. The source data underlying Figs. 1–3 are in Supplementary Data 1–3, respectively.

## Code availability

The code to replicate these analyses can be found at: https://github.com/yukamoro/GutMicrobiome_RespiratoryInfections[61].

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

## Acknowledgements

This work was supported by US National Institutes of Health under award numbers UH3OD023275, NIEHS P01ES022832, NIGMS P20GM104416, NLM R01LM012723, NLM K01LM011985, and NIGMS R01GM123014 and the US Environmental Protection Agency under award numbers RD83544201. We thank the participants and staff of the New Hampshire Birth Cohort Study for their participation and collaboration.

## Author contributions

Y.M. drafted the manuscript and conducted all statistical analyses. J.G., A.G.H., H.L. and Z.L. helped design the statistical methodology. H.G.M. conducted raw data processing. A.G.H., E.R.B., K.C.N., J.C.M., and M.R.K. contributed to designing the study, overseeing data collection and processing, and interpreting the results. All authors reviewed and revised the manuscript.

## Competing interests

The authors declare no competing interests.
