## [Peer Review File · Communications Medicine]

Reviewers' comments:

Reviewer #1 (Remarks to the Author):

Nature Communications Medicine

This is an important and interesting topic. The prospective study design and application of shotgun metagenomics in a general population are strengths.

Major concerns and issues: The data source and reliability of the diagnosis is key for this publication. 1. The source of data for diagnosis of respiratory infections, wheezing and diarrhea are unclear. In Methods, a description of parent interviews suggests that is the source. Elsewhere physician diagnosis is stated to be the source. Which is correct? Was an EMR the source? If not, that is a weakness. The paper could be significantly strengthened if physician-diagnosed, medically-attended, EMR data was the sole source of diagnosis. Even then, the criteria for diagnosis would likely vary by clinicians. At minimum, the diagnosis frequency and age when diagnosis was made and how frequently a child was counted for an infection should be described in a Table or Figure. 2. The label viral URI is presumptive since there are no viral cultures or PCR to confirm a viral infection was present. 4. The label URI implies possible viral and bacterial URI but this is presumptive since there are no bacterial cultures or PCR to confirm a bacterial respiratory infection was present. The same concern for diagnosis of LRI. Are samples available to detect pathogens by PCR? Analysis and Main Findings. The analysis of "infections and symptoms" is confusing. Were the associations and statistical tests applied to "infections" and separately to "symptoms" or was the analysis done in some other way? The findings are contradictory to what the field generally has found, i.e., reduced alpha diversity has been associated with disease. Vaginal delivery has been associated with microbiome that is associated with less disease. Supplemental Table 1 shows all outcomes worse after vaginal delivery. Therefore, contradictory to nearly all prior publications. I am concerned that the novel analysis approach may be at the root of the results conflicting with prior literature. This should be considered by the authors and a statistical reviewer should be involved. It is unclear where data presented are from 16S analysis and from shotgun metagenomics. Confounding. Mode of infant delivery is an apparent key variable. It is surprising that maternal antibiotics during pregnancy and/or antibiotic prescription for the infants is not another key variable. Methods suggest that data on antibiotic and other medications were prescribed was captured from the subjects. What was the source of that data? Why was that data not included in analysis since antibiotics would likely have an effect on microbiome? Not all antibiotics would likely have the same effect depending on their spectrum of activity. Details on maternal antibiotics and infant subject antibiotic exposure should be provided. Crisp and clear writing. Throughout the manuscript, the authors need to be more precise in their writing. Examples are provided below. Impact. The lack of any mechanistic studies linking the significant variation in bacterial species with respiratory infection susceptibility is a weakness. Discussion. The authors over reach in many of their statements (see below). Some, but not most, of the findings are linked to possible mechanistic explanations. Most paragraphs have a closing sentence suggesting need for more studies. Limitations. The limitations noted above and below are not acknowledged.

Title: "The Developing Infant Gut Microbiome and the Risk of Infection in a Prospective Birth Cohort"

The title misleads. Developing is not accurate since only a single time point of stool samples was taken. Gut is not accurate since the composition of the stool microbes included oral flora that are

not thought to be part of the gut microbiome. Infection is not accurate since the study was on respiratory infections and not all infections but did assess diarrhea also although not clear if the diarrhea was infectious and assessed wheezing which could have been viral-induced or allergic induced although not clear.

Abstract

Line 10-11. Higher infant gut microbiota alpha diversity [beta not assessed, why?] was associated with an increased risk of later [parent reported or physician-diagnosed? Upper and lower? Respiratory presumptive viral?] Infection [mostly clinically viral, no bacteria isolates] or [was it and or was it or?] respiratory symptoms, specifically [occurrence or frequency] of upper respiratory tract infections [confusing, symptoms and infections] [regardless of birth route?]] and among vaginally delivered infants with wheezing and diarrhea. [during the first year of life].

Line 12-13. Associations [which ones by both delivery routes?] were specifically observed with [oral flora Veillonella species and Haemophilus influenzae [detected in stools] among cesarean delivered infants.

Line 14-15. Our findings suggest that intestinal microbial diversity [actually the diversity arises from oral flora detected in stool samples] and the relative abundance of key taxa in early infancy may influence susceptibility to infection [what about diarrhea and wheezing] and provide opportunities for interventions to improve lifelong health [data not strong enough to add anything about interventions to conclusion]

Introduction.

Line 22. The impacts of perturbing these intricate dysbiotic [delete dysbiotic] relationships are evident in high-risk infants.

Line 23. For example, among infants [check ref if infants] with cystic fibrosis, the composition of the gut microbiome is a determinant of colonization with opportunistic pathobionts. 6,7

Line 25. Likewise in preterm infants, the gut microbiome can predict [check "predict"] potentially fatal occurrences of necrotizing enterocolitis and infection. 2,8,9

Line 30. Encouraging probiotic trials suggest health benefits from altering the gut microbiome, including enhanced immune response to pathogens²³ [for balance cite other studies have been less encouraging]

Line 37-43. Same comments as noted in abstract.

Results

Line 50. Samples collected at approximately six weeks of age [single time point should be noted in abstract and introduction]

Line 51. After removing infants for whom health information was unavailable [where was the health information obtained? The EMR?]

Line 59. Alpha diversity was positively associated with the occurrence of any infections [not precise since the study did not identify skin/soft tissue, genitourinary, etc. infections] or symptoms [not precise since the study did not identify symptoms of skin/soft tissue, genitourinary, etc. infections], and upper RTI outcomes were specifically identified [associated rather than identified]. [Need to make it clear that the study design did not include cultures or PCR to identify etiology so using the term "viral" for example, is imprecise.

Line 61. Each doubling in alpha diversity was associated with a 39% increase [not sure it is correct to state the statistic as such] in the total number of infections [what is defined as an infection? and from what source?] and symptoms [what is defined as a symptom and from what source] (RR = 1.39, 95% CI: 1.1-1.77) and a 40% increase in upper RTIs (RR = 1.40, 95% CI: 1.12-1.76)

Line 73. In GEE models of metagenomics species data, the relative abundance of Veillonella unclassified was positively associated with the total number of infections [imprecise] and respiratory symptoms [separately for infections and symptoms?] [Were all infections combined?] [Were all symptoms combined?]

Line 75. In examining specific outcomes, we found that diarrhea [no cultures or PCR, was this an infection or other causes of loose stools in infants?] was positively associated with the relative abundance of [oral flora] Streptococcus peroris and negatively associated with the relative abundance of [oral flora] Streptococcus salivarius (Fig. 78 3). [confirm only diarrhea was found in this association analysis, not URTI or LRTI or wheezing?]

Line 79. Stratified by delivery mode, we found that the total number of infections [imprecise] and symptoms [imprecise] were positively associated with [oral flora] Haemophilus influenzae among cesarean-delivered infants (RR = 81 1.02; 95% CI: 1.01-1.04) (Fig. 2C). [Separately for infections and symptoms? Were all infections combined? Were all symptoms combined?]

Line 81. [Oral flora]Veillonella parvula, Corynebacterium pseudodiphtheriticum, and Corynebacterium pseudodiphtheriticum were positively associated, while Clostridium butyricum and Coprobacillus unclassified were negatively associated, with a risk of upper RTIs among infants delivered by cesarean section (Fig. 3). [Separately for infections and symptoms? Were all infections combined? Were all symptoms combined?]

Line 89. Higher diversity of the early infant gut microbiome was associated with a greater number of infections and symptoms over the first year of life. [The sentence not needed after clarity added to prior sentences.]

Discussion

Line 87. In our prospective study of infants from the general population [in New Hampshire USA]. This is a single site with a socio-demographic that may or may not be representative of the USA or other countries.

Line 87. We observed patterns [associations] of the early developing not developing [developing to me implies multiple time points of measurement] [precisely: increased alpha diversity in the microbiome detected in stools measured at child age 6 weeks old] being related to [with] later [delete later] occurrence of infant [respiratory] infections and diarrhea and symptoms of [respiratory] infection. [during the first year of life] [nothing stated about wheezing]

Line 93. Using metagenomic sequencing, we found that Veillonella [in x delivery mode] and Haemophilus [in y delivery mode] were among the species associated with an increased risk of infant respiratory infections and symptoms.

Line 98. In our study, Veillonella, specifically Veillonella parvula, was positively associated with upper respiratory infections, especially [delete especially] in cesarean-delivered infants. Moreover, it should be noted that Veillonella parvula is an oral flora, not a typical gut flora. So would that not suggest that oral flora rather than gut flora are associate with the outcomes assessed? Why would an oral microbe be present more often in C-section delivery babies?

Line 105. While speculative, it is possible that Veillonella parvula impacts the immune response differently depending on the microbial milieu of the gut, which is modified by cesarean delivery. Further experimental studies may help to clarify this. [There is no basis in the data or the literature for this speculation. Consider removing.]

Line 109. Among cesarean-delivered infants in our study, a higher relative abundance of Corynebacterium species was associated with a greater risk of upper RTIs. Notably, not against LRTIs? Again, as the authors note Corynebacterium is an oral flora, not a typical gut flora. So would that not suggest that oral flora rather than gut flora are associate with the outcomes assessed? Why would an oral microbe be present more often in C-section delivery babies?

Line 123. Paragraph beginning “Other observations... is unbalanced since most data in the field finds reduced alpha diversity associated with disease.

Line 141 Paragraph beginning “Randomized clinical trials... should be deleted. It is unbalanced and the study did not assess any aspect of probiotic treatment nor does the study clearly point to a probiotic product to be tested.

Line 155. Paragraph beginning “Wheezing is a respiratory symptom... correctly points to the imprecision of the symptom wheezing which includes patients of different clinical and immunologic endotypes. The Discussion is unbalanced since most data in the field finds reduced alpha diversity associated with asthma. Moreover, diagnosis of asthma is challenging during infancy.

Line 162. Paragraph beginning “Diarrhea” is a symptom, not defined in Methods, that has a hodgepodge of etiologies. Most data in the field finds reduced alpha diversity associated with diarrhea associated with disease.

Line 171. Paragraph beginning “We observed that an increased relative abundance of *Streptococcus peroris* and a reduced relative abundance of *Streptococcus salivarius* were associated with a higher risk of diarrhea” includes citations and discussion of prior publications with few subjects and/or studies in mouse models that may not be applicable.

Line 186. In limitations, “we performed repeated measurements of infection occurrences [is it precise to say repeated measurements?], respiratory symptoms, and diarrhea and a broad range of potential confounding variables [what were those variables?] for the analyses.

Line 188. “A major challenge to the analysis of microbiome sequencing data is the ability to fully capture their correlated, compositional, and high-dimensional nature when assessing longitudinal outcomes. Therefore, we performed our analyses on the relative abundance of individual species and corrected for multiple hypotheses using the FDR. [This reviewer lacks the statistical expertise to assess if this is acceptable]

Line 192. “Participant recall is a potential source of bias; however, efforts were made to reduce misclassification by including questions on the duration and severity of illness [It is unclear and not stated in Methods how the investigators dealt with data on duration or severity of illness to avoid “misclassification”, and limiting our analyses to outcomes that involved either a physician visit or a prescription medication” [If the study design limited analyses to a physician visit or prescription medication then why use the parent report at all? Why not use the EMR for diagnosis since that would strengthen the paper. In pediatrics, it is unusual to prescribe antibiotics or any medication without examination of the child. Therefore, if prescriptions were given then the data of what was prescribed specifically and for what diagnosis should be available in the EMR.]

Line 194. “In a review of infants’ pediatric medical records, we found caregiver responses to be at least 80% concordant with physician assessments documented in the medical record (unpublished data).” This is puzzling to me. Why use the parent report and not the EMR? 80% concordance is good but not as good as the EMR. The concordance data should be included as supplemental if the authors cannot revise by using only EMR data.

Page 199. Conclusion paragraph should be completely rewritten to reflect the specific findings.

Methods

Line 222. Telephone interviews were conducted with infants’ caregivers quarterly in the first year of life, [key factor in validity]

Line 223. i.e., when infants turned approximately 4 months, 8 months, and 12 months of age. [3 key time points]

Line 224, Caregivers were asked whether their child had any upper RTIs or associated symptoms (e.g., runny nose, stuffy nose, eye infection, ear infection, influenza, sinus infection, pharyngitis, or laryngitis) [let me see the data to validate since sinus infection, pharyngitis not likely]

Line 226. Lower RTIs (e.g., bronchitis, pneumonia, bronchiolitis (including respiratory syncytial virus (RSV)), or whooping cough), acute respiratory symptoms (e.g., difficulty breathing, wheezing, 228 fever, or cough), or diarrhea since the previous interview. [Need data]

Line 229. For each positive response, participants were then asked whether the condition lasted more than 2 days [data], whether the child saw a physician [data], and whether the child received any prescription medications for the condition [need data]

Line 251. For our analyses, we examined the total number of reported outcomes, specifically upper RTIs and lower RTIs, as well as symptoms such as wheezing with a reported visit to a physician [not only those with a physician visit?] and treatment with a prescription medication [not only if treatment with a prescription medication] [which prescriptions, since steroids might be important to know, especially for wheezing].

Line 225. We imputed missing outcomes if the caregiver completed the interview but a specific question was unanswered using multiple imputation by chained equations and the predictive mean matching method. [How often were outcomes imputed?]

Line 266. Factors [what factors] associated with both the gut microbiome and health outcomes were considered potential confounders and included in all GEE analyses.

Line 276. Of the 465 participants included in the 16S analyses, 391 participants (84.1%) had complete data for all potentially confounding variables. [Assess missing data?] [Which confounding variables?]

Table 1, impact of missing data?

Figure 1. Are these the confounders? Maternal BMI, delivery type, sex, breast feeding at six weeks, perinatal antibiotic use, and gestational age. What about infant antibiotic exposure? That is a key confounder.

Line 452. Figure 1. Relative risk estimates represent an increased risk of infections or symptoms of infection per doubling of the inverse Simpson index. [Is it and or is it or?]

Line 453. Upper RTI, lower RTI, and wheezing outcomes are those diagnosed by a physician [must be physician diagnosed? EMR? For which a medication was prescribed [must have medication prescribed? Which medications?}. Diarrhea outcomes are those diagnosed by a physician for which no medication was prescribed [only those cases?]

Supplemental Table 1. Labels of columns and rows should be made more precise and footnotes added to further improve precision.

Reviewer #2 (Remarks to the Author):

The current study is a prospective birth cohort study that investigated the developing infant gut microbiome and risk of infection (e.g., respiratory tract infection). There are several strengths of this study, including well-characterized cohort, big sample size, use of metagenomics sequencing. I have several comments:

It is unclear that at what age the outcomes were defined.

RTI is not defined in the first place occurred.

It is a little unexpected to see *Haemophilus influenzae* show a significant association with the number of infections in the gut microbiome. This species is usually present with high abundance in

the airway. This needs to be discussed in the paper.

It is important to show the relative abundance of the top 20 genera (16S data) and species (metagenomics data) in the stool sample from this study. This will help us to identify whether *Haemophilus influenzae* is also high in the gut.

The functional capacity of the metagenomics data is not explored. This study lacks information on biological mechanisms.

The writing needs improvement in terms of logistics. From the figures, it seems the focus of the analysis is about the stratification of delivery mode, however, from the title and introduction section, I don't see those to be mentioned.

Figure 3: why there is no vaginal delivery?

Reviewer #3 (Remarks to the Author):

The manuscript reported a large-scale prospective study on the occurrence of infections and associated symptoms during the first year of life with regard to infant gut microbiome using 16S rRNA and shotgun metagenomics. The key finding is that higher alpha diversity was associated with an increased risk of later infection or respiratory symptom.

My main comment is to add some descriptive analysis for the microbiome data before conducting GEE analysis. The infant gut microbiome is fast evolving in the first year. Based on the method, the samples were collected at regularly ~6-week postpartum. It will be interesting to describe or comment on the trajectories or dynamics of the diversity profiles (at different taxonomy levels, or compare between different delivery modes). Many of those important factors are merely "controlled for" in the GEE analysis, but not explicitly analyzed and described (at least not clear to this reviewer). They can provide more meaningful context in the Discussion section.

From the perspective of review and reproducible analysis, the results provided for such large study are quite at high-level. Many tools (DADA2-phyloseq, metaphlan) described in the Method section provide rich outputs which are informative and can be provided as supplementary material to help reviewers as well as other researchers. For instance, a PCoA plot could potentially reveal important patterns.

Other minor comments:

L60: RTI should give full names spell

Response to Reviewers

We thank the reviewer for their feedback. Below, we provide a point-by-point response.

Reviewers' comments:

Reviewer #1 (Remarks to the Author):

This is an important and interesting topic. The prospective study design and application of shotgun metagenomics in a general population are strengths.

Major concerns and issues: The data source and reliability of the diagnosis is key for this publication. 1. The source of data for diagnosis of respiratory infections, wheezing and diarrhea are unclear. In Methods, a description of parent interviews suggests that is the source. Elsewhere physician diagnosis is stated to be the source. Which is correct? Was an EMR the source? If not, that is a weakness. The paper could be significantly strengthened if physician-diagnosed, medically-attended, EMR data was the sole source of diagnosis. Even then, the criteria for diagnosis would likely vary by clinicians. At minimum, the diagnosis frequency and age when diagnosis was made and how frequently a child was counted for an infection should be described in a Table or Figure.

To address this concern we have clarified our outcome ascertainment methods and provide information about the reliability of this assessment. We outline the source of our outcomes data in the Methods section, Ascertainment of Infant Health Outcomes subsection. We asked caregivers whether the child had any respiratory infections, wheezing, and diarrhea during interval phone interviews. If the caregiver gave a positive response, we further asked if these were lasted more than 2 days, whether the child saw a physician, and whether the child received any prescription medications for the condition. For our analyses, we used outcomes that were reportedly diagnosed by a physician and received prescription medication for, except for diarrhea outcomes which were reportedly diagnosed by a physician. Below we show the reliability of care giver report of physician diagnosed conditions and conditions treated with prescription medicines as compared to the medical record.

Agreement Between Caregiver Reported Outcomes and Medical Records

	All Events	Diagnosed by Physician	Medication Prescribed
All Events	80%	74%	81%
Upper Respiratory Infection	79%	74%	80%
Lower Respiratory Infection	93%	93%	95%

RSV	97%	97%	
Respiratory Symptoms	49%	63%	84%
Wheezing	89%	92%	95%
Diarrhea	67%	83%	97%

As shown, the reliability is generally high. Nonetheless, in response to this comment and the comment below, we have included the following limitation in lines 250 – 253:

“Our outcomes were not ascertained by viral or bacterial culture or PCR to confirm type of infection; as a result, we relied on responses of telephone surveys given to caregivers.” We also mention the concordance between our caregiver responses and medical records as a limitation in lines 256 - 258:

“In a review of infants’ pediatric medical records, we found caregiver responses to be at least 80% concordant with physician assessments documented in the medical record (unpublished data)”

We further included the number of each outcome at each age in our new Supplementary Table 1 and Supplementary Table 2. We also refer to these table in our new sentence in the Results section in lines 68 - 70:

“The numbers of each respiratory infection and symptom at each age is provided in Supplementary Table 1 and Supplementary Table 2.”

2. The label viral URI is presumptive since there are no viral cultures or PCR to confirm a viral infection was present. 4. The label URI implies possible viral and bacterial URI but this is presumptive since there are no bacterial cultures or PCR to confirm a bacterial respiratory infection was present. The same concern for diagnosis of LRI. Are samples available to detect pathogens by PCR?

We were unable to perform a viral/bacterial culture or PCR to confirm the type of URI. Thus, we do not label our infection outcomes as bacterial, viral, or fungal. As mentioned above, we now include this as a limitation in lines 250 – 253:

“Our outcomes were not ascertained by viral or bacterial culture or PCR to confirm type of infection; as a result, we relied on responses of telephone surveys given to caregivers.”

Analysis and Main Findings. The analysis of “infections and symptoms” is confusing. Were the associations and statistical tests applied to “infections” and separately to “symptoms” or was the analysis done in some other way?

To address this concern, we have clarified our approach. “Any infections and symptoms” refer to the following outcomes: upper RTIs or associated symptoms (e.g., runny nose, stuffy nose, eye infection, ear infection, influenza, sinus infection, pharyngitis, or laryngitis), lower RTIs (e.g., bronchitis, pneumonia, bronchiolitis (including respiratory syncytial virus (RSV)), or whooping cough), and acute respiratory

symptoms (e.g., difficulty breathing, wheezing, fever, or cough). We summed the total number of the aforementioned outcomes to form this variable. We have made this clearer in the first sentence of the Results section, 16S V4-V5 rRNA Gene: Alpha Diversity subsection, lines 73 - 76:

“Associations were determined via Wald test in generalized estimating equation (GEE) analyses with a p-value threshold of 0.05. Alpha diversity was positively associated with the occurrence of any infections or symptoms, which include upper respiratory tract infections (RTI), lower RTI, and acute respiratory symptoms.”

The findings are contradictory to what the field generally has found, i.e., reduced alpha diversity has been associated with disease. Vaginal delivery has been associated with microbiome that is associated with less disease. Supplemental Table 1 shows all outcomes worse after vaginal delivery. Therefore, contradictory to nearly all prior publications. I am concerned that the novel analysis approach may be at the root of the results conflicting with prior literature. This should be considered by the authors and a statistical reviewer should be involved.

We acknowledge that the findings for alpha diversity are contradictory to some prior publications. Though this may be the case, we are looking at an early microbiome when diversity is low in healthy babies. Thus, higher diversity very early in life may not an indicator of better health. For example, higher diversity is initially seen in formula fed infants¹, but their outcomes are noted to be poorer. Many studies have shown no associations between alpha diversity and adverse health outcomes, but have demonstrated differences in abundance of specific bacteria². For, these reasons, we used our metagenomics data to look at specific microbial species in relation to our outcomes of interest. We added the following sentences in the discussion section to address this concern in lines 18 - 189:

“Although our findings for alpha diversity may seem contradictory to some prior studies that reported negative associations between alpha diversity and health outcomes⁴³, our study focuses on the early microbiome when diversity is low in healthy babies. We further prospectively examined associations with respiratory infections. Similar to our study, other studies found no associations between alpha diversity and adverse health outcomes, but observed differences in abundance of specific microbes.”

Furthermore, we wish to clarify that our stratified analyses indicate associations within the two delivery modes that may be related to increased or decreased risk of respiratory infections and symptoms.

1. Ma, J. *et al.* Comparison of gut microbiota in exclusively breast-fed and formula-fed babies: a study of 91 term infants. *Sci Rep* **10**, 15792 (2020).
2. Tamburini, S., Shen, N., Wu, H. C. & Clemente, J. C. The microbiome in early life: implications for health outcomes. *Nat Med* **22**, 713–722 (2016).

It is unclear where data presented are from 16S analysis and from shotgun metagenomics.

We now make clearer that our analyses on alpha diversity are conducted on 16S data. Our analyses with species are conducted on metagenomics data. The subtitles in the results section provide this information, and further description is provided in the Methods section, Statistical Analysis subsection, paragraph 2.

Confounding. Mode of infant delivery is an apparent key variable. It is surprising that maternal antibiotics during pregnancy and/or antibiotic prescription for the infants is not another key variable. Methods suggest that data on antibiotic and other medications were prescribed was captured from the subjects. What was the source of that data? Why was that data not included in analysis since antibiotics would likely have an effect on microbiome? Not all antibiotics would likely have the same effect depending on their spectrum of activity. Details on maternal antibiotics and infant subject antibiotic exposure should be provided.

We describe in our Methods section that maternal prenatal antibiotic use was reported in prenatal records. We include maternal antibiotic use during pregnancy as a confounder in our analyses as mentioned in line 339. We did not adjust for infant antibiotics use they may be used to treat our outcomes of interest. Because antibiotics is a consequence of the outcome, we do not adjust for antibiotics and other medications prescribed to the infant.

Crisp and clear writing. Throughout the manuscript, the authors need to be more precise in their writing. Examples are provided below. Impact. The lack of any mechanistic studies linking the significant variation in bacterial species with respiratory infection susceptibility is a weakness. Discussion. The authors over reach in many of their statements (see below). Some, but not most, of the findings are linked to possible mechanistic explanations. Most paragraphs have a closing sentence suggesting need for more studies. Limitations. The limitations noted above and below are not acknowledged.

Title: "The Developing Infant Gut Microbiome and the Risk of Infection in a Prospective Birth Cohort"

The title misleads. Developing is not accurate since only a single time point of stool samples was taken. Gut is not accurate since the composition of the stool microbes included oral flora that are not thought to be part of the gut microbiome. Infection is not accurate since the study was on respiratory infections and not all infections but did assess diarrhea also although not clear if the diarrhea was infectious and assessed wheezing which could have been viral-induced or allergic induced although not clear.

We appreciate the reviewers suggestions to make our manuscript more precise. We changed the title of the manuscript to reflect our findings more accurately:

"The infant stool microbiome in relation to subsequent risk of respiratory infections and symptoms among vaginally and cesarean delivered infants"

Abstract

Line 10-11. Higher infant gut microbiota alpha diversity [beta not assessed, why?] was associated with an increased risk of later [parent reported or physician-diagnosed? Upper and lower? Respiratory presumptive viral?] Infection [mostly clinically viral, no bacteria isolates] or [was it and or was it or?]

respiratory symptoms, specifically [occurrence or frequency] of upper respiratory tract infections [confusing, symptoms and infections] [regardless of birth route?]] and among vaginally delivered infants with wheezing and diarrhea. [during the first year of life].

We appreciate the reviewer's points. We now clarify the method of outcome ascertainment, and clarify which outcomes were assessed and in which subgroup and time period. We were unable to evaluate beta diversity due to the longitudinal nature of our outcome data and the lack of statistical methods for such data.

Line 12-13. Associations [which ones by both delivery routes?] were specifically observed with [oral flora Veillonella species and Haemophilus influenzae [detected in stools] among cesarean delivered infants.

Cesarian delivered infants did not have fecal or vaginal exposure at birth, so their primary exposure is postnatally: e.g. through breast milk, skin, and other sources. We have clarified the delivery routes in the abstract. We now address the issue regarding species that are found in the oral flora in the discussion section. We also included text to make the mode of delivery clear in lines 13 - 14:

"Associations were specifically observed with Veillonella species among all deliveries and Haemophilus influenzae among cesarean-delivered infants."

We address the issue of species found in the oral flora in the discussion section in lines 124 – 136:

"Our analyses found associations with many bacterial species that are commonly found in oral flora, though they also have been detected in the gut. An early driver of the gut microbiome is diet. Infants born operatively may be relatively more likely to acquire the genera of such species through breast milk²⁶."

Line 14-15. Our findings suggest that intestinal microbial diversity [actually the diversity arises from oral flora detected in stool samples] and the relative abundance of key taxa in early infancy may influence susceptibility to infection [what about diarrhea and wheezing] and provide opportunities for interventions to improve lifelong health [data not strong enough to add anything about interventions to conclusion]

To clarify, we changed text to the last sentence of the abstract to include wheeze and diarrhea and remove the phrase regarding interventions in line 16:

"Our findings suggest that intestinal microbial diversity and the relative abundance of key taxa in early infancy may influence susceptibility to infection, wheezing, and diarrhea.

Introduction.

Line 22. The impacts of perturbing these intricate dysbiotic [delete dysbiotic] relationships are evident in high-risk infants.

We removed "dysbiotic" from the sentence in line 27.

Line 23. For example, among infants [check ref if infants] with cystic fibrosis, the composition of the gut microbiome is a determinant of colonization with opportunistic pathobionts. 6,7

We have checked our citations and confirmed the studies are conducted on infant microbiomes.

Line 25. Likewise in preterm infants, the gut microbiome can predict [check “predict”] potentially fatal occurrences of necrotizing enterocolitis and infection. 2,8,9

We changed the word “predict” to “is associated with” in line 30.

Line 30. Encouraging probiotic trials suggest health benefits from altering the gut microbiome, including enhanced immune response to pathogens²³ [for balance cite other studies have been less encouraging]

We added another review article that discusses the controversy in the benefits of probiotics on health.

“Encouraging probiotic trials suggest health benefits from altering the gut microbiome, including enhanced immune response to pathogens^{23,24}”

24 Tamburini, S., Shen, N., Wu, H. C. & Clemente, J. C. The microbiome in early life: implications for health outcomes. *Nat Med* **22**, 713–722 (2016).

Line 37-43. Same comments as noted in abstract.

We have clarified the delivery routes in the introduction in lines 48 – 51:

“Using next-generation sequencing (NGS), shotgun metagenomics, *Veillonella* in all deliveries and *Haemophilus* in cesarean deliveries were among the species in six-week stool identified as being related to an increased risk of subsequent respiratory infections in infants’ first year of life.”

We also address the issue of species found in the oral flora in the discussion section lines 124 – 136:

“Our analyses found associations with many bacterial species that are commonly found in oral flora, though they also have been detected in the gut. An early driver of the gut microbiome is diet. Infants born operatively may be relatively more likely to acquire the genera of such species through breast milk.”

Results

Line 50. Samples collected at approximately six weeks of age [single time point should be noted in abstract and introduction]

We now include this information in the second sentence of our abstract in line 9:

“We examined the occurrence of infections and associated symptoms during the first year of life in relation to infant gut microbiome at six weeks of age using bacterial 16S rRNA V4-V5 gene sequencing (n = 465) and shotgun metagenomics (n = 185).”

We also added this information to the last two sentences in the introduction paragraph in lines 42-51:

“Based on amplicon sequence variant (ASV) data generated from 16S rRNA sequencing, higher gut microbiome diversity at six weeks of age was associated with an increased risk of infection requiring prescription medicines or symptoms of infection involving a visit to a health care provider, with associations varying by delivery mode. Using next-generation sequencing (NGS), shotgun metagenomics, *Veillonella* and *Haemophilus* were among the species in six-week stool identified as being related to an increased risk of subsequent respiratory infections in infants’ first year of life.”

Line 51. After removing infants for whom health information was unavailable [where was the health information obtained? The EMR?]

We now clarify this in the Results section, Baseline Characteristics subsection in line 59:

“After removing infants for whom health information was unavailable in telephone surveys, ...”

Line 59. Alpha diversity was positively associated with the occurrence of any infections [not precise since the study did not identify skin/soft tissue, genitourinary, etc. infections] or symptoms [not precise since the study did not identify symptoms of skin/soft tissue, genitourinary, etc. infections], and upper RTI outcomes were specifically identified [associated rather than identified]. [Need to make it clear that the study design did not include cultures or PCR to identify etiology so using the term “viral” for example, is imprecise.

We now include clarification of our definition of “all infections and symptoms” in the first sentence of the Results section, 16S V4-V5 rRNA Gene: Alpha Diversity subsection in lines 74 – 76:

“Alpha diversity was positively associated with the occurrence of any respiratory infection or symptom of infection, which include upper respiratory tract infections (RTI), lower RTI, and acute respiratory symptoms.”

We changed the wording in line 77 to “associated” instead of “identified”. We made it clear in our limitations section that we did not include cultures or PCR to identify etiology in lines 250 - 252:

“Our outcomes were not ascertained by viral or bacterial culture or PCR to confirm infection; as a result, we relied on responses of telephone surveys given to caregivers.”

Line 61. Each doubling in alpha diversity was associated with a 39% increase [not sure it is correct to state the statistic as such] in the total number of infections [what is defined as an infection? and from what source?] and symptoms [what is defined as a symptom and from what source] (RR = 1.39, 95% CI: 1.1-1.77) and a 40% increase in upper RTIs (RR = 1.40, 95% CI: 1.12-1.76)

We now clarify our definition of “all infections and symptoms” in lines 74 – 76:

“Alpha diversity was positively associated with the occurrence of any respiratory infection or symptom of infection, which include upper respiratory tract infections (RTI), lower RTI, and acute respiratory symptoms.”

We also clarify the interpretation of our results in the results section and figure legend of Figure 1, as well as in the table legend of Supplementary Table 3.

We clarify the interpretation of our alpha diversity results in lines 76 to 82:

“Upper RTI outcomes were specifically associated. Each doubling in alpha diversity was associated with a 39% increase in having an additional infection or symptom of infection (RR = 1.39, 95% CI: 1.1-1.77) and a 40% increase in an additional upper RTI (RR = 1.40, 95% CI: 1.12-1.76) (Fig. 1, Supplementary Table 3). Among vaginally delivered infants, a doubling of alpha diversity was associated with a 62% increase in having an additional infection or symptom of infection (RR = 1.62, 95% CI: 1.23-2.15) (Fig. 1, Supplementary Table 3).”

Figure 1 clarification:

“Relative risk estimates represent an increased risk of having an additional infection or symptom of infection or an increased risk of experiencing wheezing or diarrhea with each doubling of the inverse Simpson index.”

Supplementary Table 3 clarification:

“Relative risk estimates represent an increased risk of having an additional infection or symptom of infection or an increased risk of experiencing wheezing or diarrhea with each doubling of the inverse Simpson index.”

The source is telephone surveys as outlined in our Methods section.

Line 73. In GEE models of metagenomics species data, the relative abundance of *Veillonella* unclassified was positively associated with the total number of infections [imprecise] and respiratory symptoms [separately for infections and symptoms?] [Were all infections combined?] [Were all symptoms combined?]

We included clarification of our definition of “all infections and symptoms” in the first sentence of the Results section, 16S V4-V5 rRNA Gene: Alpha Diversity subsection in lines 74 – 76:

“Alpha diversity was positively associated with the occurrence of any respiratory infection or symptom of infection, which include upper respiratory tract infections (RTI), lower RTI, and acute respiratory symptoms.”

We also clarify the interpretation of our species-level results in the results section and figure legends of Figure 2 and Figure 3.

We clarify the interpretation of our metagenomics species results in lines 89 to 91 and lines 109 to 111:

“In GEE of metagenomics species data, the doubling of relative abundance of *Veillonella* unclassified was positively associated with having an additional infection or symptom of infection in the first year of life (RR = 1.02; 95% CI: 1.01-1.04) (Fig. 2a).”

“Stratified by delivery mode, we found that having an additional infection or symptom of infection was positively associated with *Haemophilus influenzae* among cesarean-delivered infants (RR = 1.02; 95% CI: 1.01-1.04) (Fig. 2C).”

Figure 2 clarification:

“Relative risk estimates represent an increased risk of having an additional infections or symptoms of infection with each doubling of relative abundance.”

Figure 3 clarification:

“Relative risk estimates represent an increased risk of having an additional upper respiratory infection or an increased risk of experiencing diarrhea with each doubling of the relative abundance.”

Line 75. In examining specific outcomes, we found that diarrhea [no cultures or PCR, was this an infection or other causes of loose stools in infants?] was positively associated with the relative abundance of [oral flora] *Streptococcus peroris* and negatively associated with the relative abundance of [oral flora] *Streptococcus salivarius* (Fig. 78 3). [confirm only diarrhea was found in this association analysis, not URTI or LRTI or wheezing?]

As indicated, we now make clear in our limitations section that we did not include cultures or PCR to identify etiology in lines 206 - 208:

“Our outcomes were not ascertained by viral or bacterial culture or PCR to confirm infection; as a result, we relied on responses of telephone surveys given to caregivers.”

Line 79. Stratified by delivery mode, we found that the total number of infections [imprecise] and symptoms [imprecise] were positively associated with [oral flora] *Haemophilus influenzae* among cesarean-delivered infants (RR = 1.02; 95% CI: 1.01-1.04) (Fig. 2C). [Separately for infections and symptoms? Were all infections combined? Were all symptoms combined?]

We now clarify that the outcome “any infection and symptoms” are total combined counts of infections and symptoms reported in a given time period.

We included clarification of our definition of “any infections and symptoms” in the results section in lines 68 to 70:

“Alpha diversity was positively associated with the occurrence of any respiratory infection or symptom of infection, which include upper respiratory tract infections (RTI), lower RTI, and acute respiratory symptoms.”

Line 81. [Oral flora]*Veillonella parvula*, *Corynebacterium pseudodiphtheriticum*, and *Corynebacterium*

pseudodiphtheriticum were positively associated, while Clostridium butyricum and Coprobacillus unclassified were negatively associated, with a risk of upper RTIs among infants delivered by cesarean section (Fig. 3). [Separately for infections and symptoms? Were all infections combined? Were all symptoms combined?]

We now clarify that the bacterial species were associated with one additional upper RTI. We included this clarification in lines 93-94:

“...were negatively associated, with a risk of an additional upper RTI among infants delivered by cesarean section (Fig. 3).”

We also address the issue of species found in the oral flora in the discussion section as noted in our previous response.

Line 89. Higher diversity of the early infant gut microbiome was associated with a greater number of infections and symptoms over the first year of life. [The sentence not needed after clarity added to prior sentences.]

Clarity has been added to aforementioned sentences to describe infections and symptoms as noted in our previous responses.

Discussion

Line 87. In our prospective study of infants from the general population [in New Hampshire USA]. This is a single site with a socio-demographic that may or may not be representative of the USA or other countries.

We added “in New Hampshire, USA” in line 97 of the main text.

Line 87. We observed patterns [associations] of the early developing not developing [developing to me implies multiple time points of measurement] [precisely: increased alpha diversity in the microbiome detected in stools measured at child age 6 weeks old] being related to [with] later [delete later] occurrence of infant [respiratory] infections and diarrhea and symptoms of [respiratory] infection. [during the first year of life] [nothing stated about wheezing]

We made the following edits to address concerns with the sentence:

“In our prospective study of infants from the general population in New Hampshire, USA, we observed patterns of the early microbiome that were related to the occurrence of infant respiratory infections, wheezing, and diarrhea”

Line 93. Using metagenomic sequencing, we found that Veillonella [in x delivery mode] and Haemophilus [in y delivery mode] were among the species associated with an increased risk of infant respiratory infections and symptoms.

We now specify the mode of delivery in lines 103-105 of the main text:

“Using metagenomic sequencing, we found that *Veillonella* in any delivery mode and *Haemophilus* in cesarean deliveries were among the species associated with an increased risk of infant respiratory infections and symptoms.”

Line 98. In our study, *Veillonella*, specifically *Veillonella parvula*, was positively associated with upper respiratory infections, especially [delete especially] in cesarean-delivered infants. Moreover, it should be noted that *Veillonella parvula* is an oral flora, not a typical gut flora. So would that not suggest that oral flora rather than gut flora are associated with the outcomes assessed? Why would an oral microbe be present more often in C-section delivery babies?

The reviewer brings up a very interesting point. We now clarify that bacteria commonly found in oral flora may have originated from other body sites or the environment. For instance *Veillonella parvula* are also observed in both gut and oral ecosystems¹.

We also note that the vast majority of our infants were breast fed, an important driver of the infant gut microbiome, and these species may be found in breast milk². This may be especially important among Caesarian born infants who are not exposed to maternal fecal or vaginal microbes during delivery. We address the issue of species found in the oral flora in the discussion section in lines 105 – 109:

“Our analyses found associations with many bacterial species that are commonly found in oral flora, though they also have been detected in the gut. An early driver of the gut microbiome is diet. Infants born operatively may be relatively more likely to acquire the genera of such species through breast milk²⁶”

1. Poppleton DI, Duchateau M, Hourdel V, Matondo M, Flechsler J, Klingl A, Beloin C, Gribaldo S. Outer Membrane Proteome of *Veillonella parvula*: A Diderm Firmicute of the Human Microbiome. *Front Microbiol.* 2017 Jun 30;8:1215. doi: 10.3389/fmicb.2017.01215.

2. Lundgren, S. N. *et al.* Maternal diet during pregnancy is related with the infant stool microbiome in a delivery mode-dependent manner. *Microbiome* 6, 109 (2018).

Line 105. While speculative, it is possible that *Veillonella parvula* impacts the immune response differently depending on the microbial milieu of the gut, which is modified by cesarean delivery. Further experimental studies may help to clarify this. [There is no basis in the data or the literature for this speculation. Consider removing.]

Thank you for the suggestion; we have removed the sentences.

Line 109. Among cesarean-delivered infants in our study, a higher relative abundance of *Corynebacterium* species was associated with a greater risk of upper RTIs. Notably, not against LRTIs? Again, as the authors note *Corynebacterium* is an oral flora, not a typical gut flora. So would that not suggest that oral flora rather than gut flora are associated with the outcomes assessed? Why would an oral microbe be present more often in C-section delivery babies?

We have elaborated on issue of species found in the oral flora in the discussion section in lines 105 – 109:

“Our analyses found associations with many bacterial species that are commonly found in oral flora, though they also have been detected in the gut. An early driver of the gut microbiome is diet. Infants born operatively may be relatively more likely to acquire the genera of such species through breast milk”

Line 123. Paragraph beginning “Other observations... is unbalanced since most data in the field finds reduced alpha diversity associated with disease.

We added a sentence to this paragraph in lines 142 to 143 to clarify:

“Other studies have also found associations between the gut microbiome and respiratory infections.”

Line 141 Paragraph beginning “Randomized clinical trials... should be deleted. It is unbalanced and the study did not assess any aspect of probiotic treatment nor does the study clearly point to a probiotic product to be tested.

Thank you for your suggestion; we have removed the paragraph.

Line 155. Paragraph beginning “Wheezing is a respiratory symptom... correctly points to the imprecision of the symptom wheezing which includes patients of different clinical and immunologic endotypes. The Discussion is unbalanced since most data in the field finds reduced alpha diversity associated with asthma. Moreover, diagnosis of asthma is challenging during infancy.

We acknowledge that diagnosis of asthma is challenging during infancy; our wheezing variable does not differentiate between various causes of wheezing. Thus, our last sentence in this paragraph:

“Whether our findings translate to a later risk of asthma will require longer-term follow-up of our cohort.”

We now include this as a limitation in our limitations section of our discussion in lines 208 – 209:

“Additionally, we could not differentiate between the various causes of wheezing and diarrhea in our dataset”

Line 162. Paragraph beginning “Diarrhea” is a symptom, not defined in Methods, that has a hodgepodge of etiologies. Most data in the field finds reduced alpha diversity associated with diarrhea associated with disease.

We agree with the reviewer that diarrhea is a heterogenous outcome. We now explained in the Methods section and table/figure legends that “diarrhea” is a patient-reported outcome of diarrhea that was diagnosed by a physician and acknowledge the limitations of this outcome in the discussion in lines 208 – 209:

“Additionally, we could not differentiate between the various causes of wheezing and diarrhea in our dataset”

Line 171. Paragraph beginning “We observed that an increased relative abundance of *Streptococcus peroris* and a reduced relative abundance of *Streptococcus salivarius* were associated with a higher risk of diarrhea” includes citations and discussion of prior publications with few subjects and/or studies in mouse models that may not be applicable.

We address this concern in the last sentence of this paragraph in lines 195-196:

“Together, these findings raise the possibility of a role of *Streptococcus* species in susceptibility to diarrhea in early childhood.”

Line 186. In limitations, “we performed repeated measurements of infection occurrences [is it precise to say repeated measurements?], respiratory symptoms, and diarrhea and a broad range of potential confounding variables [what were those variables?] for the analyses.

We agree and changed the word “performed” to “examined” to make this clearer.

We also included a list of confounders to the sentence in lines 200-202.

“and a broad range of potential confounding variables, such as maternal prepregnancy BMI, delivery mode, infant sex, breast feeding at six weeks, antibiotic use during pregnancy, and gestational age, for the analyses.”

Line 188. “A major challenge to the analysis of microbiome sequencing data is the ability to fully capture their correlated, compositional, and high-dimensional nature when assessing longitudinal outcomes. Therefore, we performed our analyses on the relative abundance of individual species and corrected for multiple hypotheses using the FDR. [This reviewer lacks the statistical expertise to assess if this is acceptable]

Line 192. “Participant recall is a potential source of bias; however, efforts were made to reduce misclassification by including questions on the duration and severity of illness [It is unclear and not stated in Methods how the investigators dealt with data on duration or severity of illness to avoid “misclassification”, and limiting our analyses to outcomes that involved either a physician visit or a prescription medication” [If the study design limited analyses to a physician visit or prescription medication then why use the parent report at all? Why not use the EMR for diagnosis since that would strengthen the paper. In pediatrics, it is unusual to prescribe antibiotics or any medication without examination of the child. Therefore, if prescriptions were given then the data of what was prescribed specifically and for what diagnosis should be available in the EMR.]

Line 194. “In a review of infants’ pediatric medical records, we found caregiver responses to be at least 80% concordant with physician assessments documented in the medical record (unpublished data).”

This is puzzling to me. Why use the parent report and not the EMR? 80% concordance is good but not as good as the EMR. The concordance data should be included as supplemental if the authors cannot revise by using only EMR data.

We use outcomes reported by caregivers in telephone surveys to maximize our sample size. We are confident in these data because we did have medical records on a subset of our participants, and the concordance was relatively high with medical records. Below we summarize the concordance between caregiver response and medical records.

Agreement Between Caregiver Reported Outcomes and Medical Records

	All Events	Diagnosed by Physician	Medication Prescribed
All Events	80%	74%	81%
Upper Respiratory Infection	79%	74%	80%
Lower Respiratory Infection	93%	93%	95%
RSV	97%	97%	
Respiratory Symptoms	49%	63%	84%
Wheezing	89%	92%	95%
Diarrhea	67%	83%	97%

As shown, the reliability is generally high. We mention the high concordance between our caregiver responses and medical records in the limitation section in lines 212-214:

“In a review of infants’ pediatric medical records, we found caregiver responses to be at least 80% concordant with physician assessments documented in the medical record (unpublished data)”

Page 199. Conclusion paragraph should be completely rewritten to reflect the specific findings.

We now include our main findings in the concluding paragraph of the discussion section, in lines 265-272:

“In conclusion, our findings from a prospective birth cohort of US infants suggest that the composition of the microbiome in early life influences the most common health outcomes of infancy, which in turn may have consequences on lifelong disease risk. While higher alpha diversity was associated respiratory

infections and symptoms overall and among vaginal deliveries, the doubling of relative abundance of unclassified Veillonella species and Haemophilus influenza species increased risk of an additional respiratory infection or symptom overall and among cesarean-born infants respectively. Our findings may help to inform interventions aimed at altering the microbiome during this critical window of immune training.”

Methods

Line 222. Telephone interviews were conducted with infants’ caregivers quarterly in the first year of life, [key factor in validity]

Line 223. i.e., when infants turned approximately 4 months, 8 months, and 12 months of age. [3 key time points]

We removed the word “quarterly” from this sentence.

Line 224, Caregivers were asked whether their child had any upper RTIs or associated symptoms (e.g., runny nose, stuffy nose, eye infection, ear infection, influenza, sinus infection, pharyngitis, or laryngitis) [let me see the data to validate since sinus infection, pharyngitis not likely]

We report the total number of outcomes at each time period in our new Supplementary Figure 1 and Supplementary Figure 2. There were 2 subjects who had reported a sinus infection by 8 months of life, and 3 subjects who had reported a sinus infection by 12 months of life. There was no report of pharyngitis our dataset.

Line 226. Lower RTIs (e.g., bronchitis, pneumonia, bronchiolitis (including respiratory syncytial virus (RSV)), or whooping cough), acute respiratory symptoms (e.g., difficulty breathing, wheezing, 228 fever, or cough), or diarrhea since the previous interview. [Need data] Line 229. For each positive response, participants were then asked whether the condition lasted more than 2 days [data], whether the child saw a physician [data], and whether the child received any prescription medications for the condition [need data]

We report the total number of outcomes at each time period in our new Supplementary Figure 1 and Supplementary Figure 2. There were 2 subjects who had reported a sinus infection by 8 months of life, and 3 subjects who had reported a sinus infection by 12 months of life. There was no report of pharyngitis our dataset.

Line 251. For our analyses, we examined the total number of reported outcomes, specifically upper RTIs and lower RTIs, as well as symptoms such as wheezing with a reported visit to a physician [not only those with a physician visit?] and treatment with a prescription medication [not only if treatment with a prescription medication] [which prescriptions, since steroids might be important to know, especially for wheezing].

To clarify, outcomes of upper and lower RTI and respiratory symptoms such as wheezing are those that were reported by the caregiver as diagnosed by a physician **and** treated with prescription medication.

We do not include the type of prescription medication in our models because they are administered subsequent to occurrence of outcome.

Line 225. We imputed missing outcomes if the caregiver completed the interview but a specific question was unanswered using multiple imputation by chained equations and the predictive mean matching method. [How often were outcomes imputed?]

Only a small number of missing outcomes were imputed: 24 for 16S analyses and 7 for metagenomics analyses. This information is now included in the new Supplementary Table 1 and Supplementary Table 2.

Line 266. Factors [what factors] associated with both the gut microbiome and health outcomes were considered potential confounders and included in all GEE analyses.

The factors are discussed the following sentence starting at line 288, and we changed the word “variables” to “confounders” to make these more clear:

“These confounders included maternal prepregnancy BMI (kg/m²), delivery mode (vaginal/cesarian), infant sex (male/female), breast feeding at six weeks (exclusively breastfed/mixed fed or exclusively formula fed), antibiotic use during pregnancy (yes/no), and gestational age (completed weeks).”

Line 276. Of the 465 participants included in the 16S analyses, 391 participants (84.1%) had complete data for all potentially confounding variables. [Assess missing data?] [Which confounding variables?]

Our Table 1 shows the total number of subjects with complete data for each variable. We discuss in lines 299 - 301 that we assume missing confounder entries were missing at random:

“We assumed missing confounder entries were missing at random and used multiple imputation by chained equations and the predictive mean matching method to impute missing data.”

Table 1, impact of missing data?

We discuss how many subjects had missing data for each variable in the table. We assumed confounder entries are were missing at random, so our imputed values in our final model represent initial distributions.

Figure 1. Are these the confounders? Maternal BMI, delivery type, sex, breast feeding at six weeks, perinatal antibiotic use, and gestational age. What about infant antibiotic exposure? That is a key confounder.

Yes, we include maternal BMI, delivery type, sex, breast feeding at six weeks, perinatal antibiotic use, and gestational age in our statistical analyses as confounders as outlined in our methods section. As mentioned in our previous response, we did not adjust for infant antibiotics use they may be used to treat our outcomes of interest

Line 452. Figure 1. Relative risk estimates represent an increased risk of infections or symptoms of infection per doubling of the inverse Simpson index. [Is it and or is it or?]

We summed the number of infections **and** symptoms of infection for this variable; therefore, this variable represents any infection **or** symptom. We changed the wording of this sentence to make the interpretations more clear:

“Relative risk estimates represent an increased risk of having an additional infection or symptom of infection or an increased risk of experiencing wheezing or diarrhea with each doubling of the inverse Simpson index.”

Line 453. Upper RTI, lower RTI, and wheezing outcomes are those diagnosed by a physician [must be physician diagnosed? EMR? For which a medication was prescribed [must have medication prescribed? Which medications?]. Diarrhea outcomes are those diagnosed by a physician for which no medication was prescribed [only those cases?]

Upper RTI, lower RTI, and wheezing outcomes are those diagnosed by a physician and medication was prescribed. This is to ensure that outcomes are accurate and concordant with medical records. Diarrhea outcomes are those diagnosed by physician due to the small number of subjects who received prescription medication for diarrhea.

Supplemental Table 1. Labels of columns and rows should be made more precise and footnotes added to further improve precision.

Thank you for your suggestion. We added more clarifying information to the outcome “Any Infection or Symptom” in the table legend:

“Relative risk estimates represent an increased risk of having an additional infection or symptom of infection or an increased risk of experiencing wheezing or diarrhea with each doubling of the inverse Simpson index.”

Reviewer #2 (Remarks to the Author):

The current study is a prospective birth cohort study that investigated the developing infant gut microbiome and risk of infection (e.g., respiratory tract infection). There are several strengths of this study, including well-characterized cohort, big sample size, use of metagenomics sequencing. I have several comments:

It is unclear that at what age the outcomes were defined.

Outcomes were assessed at 4 months, 8 months, and 12 months of age as described in the methods section in line 242. We also include new Supplementary table 1 and Supplementary Table 2 to display the number of outcomes at each age.

RTI is not defined in the first place occurred.

We have corrected this and defined RTI in our main text in line 70.

It is a little unexpected to see *Haemophilus influenzae* show a significant association with the number of infections in the gut microbiome. This species is usually present with high abundance in the airway. This needs to be discussed in the paper.

To address this concern, we now discuss *Haemophilus influenzae* in our new paragraph 4 of the Discussion section, starting at lines 135 - 141:

“Caesarean-delivered infants in our study had a positive association between *Haemophilus influenzae* and number of any respiratory infections and symptoms. *Haemophilus influenzae* is a bacterial species known to cause several types of infectious diseases, including those of the respiratory tract. Although previous studies have not found associations between *Haemophilus influenzae* in the gut microbiome and respiratory infections, the species have been found to reside in the intestinal tract³⁶. Further explorations of the gut-lung axis, as well as origin of such bacteria in the gut, are warranted³⁷.”

It is important to show the relative abundance of the top 20 genera (16S data) and species (metagenomics data) in the stool sample from this study. This will help us to identify whether *Haemophilus influenzae* is also high in the gut.

Thank you for your suggestion. We now include a heat map of the most common genera and species in our new Supplementary Figure 1. *Haemophilus influenzae* was not commonly found in 6-week stool in our metagenomics data. However, *Haemophilus* was commonly found in our 16S data.

The functional capacity of the metagenomics data is not explored. This study lacks information on biological mechanisms.

We appreciate the reviewer’s concern. We are aware that the functional capacity is not explored in our analyses. We hope to investigate this in future analyses.

The writing needs improvement in terms of logistics. From the figures, it seems the focus of the analysis is about the stratification of delivery mode, however, from the title and introduction section, I don’t see those to be mentioned.

We now include more information in the title and introduction. Our new title is as follows:

“The infant stool microbiome in relation to subsequent risk of respiratory infections and symptoms among vaginally and cesarean delivered infants”

We clarify the our findings to include mode of delivery in the introduction in lines 13 – 14:

“Associations were specifically observed with *Veillonella* species among all deliveries and *Haemophilus influenzae* among Cesarean-delivered infants.”

Figure 3: why there is no vaginal delivery?

We did not include vaginal delivery in Figure 3 because we did not observe any associations that met FDR corrections. This is mentioned in the legend of Figure 3.

Reviewer #3 (Remarks to the Author):

The manuscript reported a large-scale prospective study on the occurrence of infections and associated symptoms during the first year of life with regard to infant gut microbiome using 16S rRNA and shotgun metagenomics. The key finding is that higher alpha diversity was associated with an increased risk of later infection or respiratory symptom.

My main comment is to add some descriptive analysis for the microbiome data before conducting GEE analysis. The infant gut microbiome is fast evolving in the first year. Based on the method, the samples were collected at regularly ~6-week postpartum. It will be interesting to describe or comment on the trajectories or dynamics of the diversity profiles (at different taxonomy levels, or compare between different delivery modes). Many of those important factors are merely "controlled for" in the GEE analysis, but not explicitly analyzed and described (at least not clear to this reviewer). They can provide more meaningful context in the Discussion section.

We appreciate the reviewer's concern. We now include a heat map visualizing the relative abundance of the 20 most commonly found genera and species in our 16S and metagenomics data respectively in our new Supplementary Figure 1. We also add text to our Results section to summarize common genera and species starting at line 58:

“The five most common genera in our 16S data were *Escherichia/Shigella*, *Bacteroides*, *Bifidobacterium*, *Klebsiella*, and *Enterococcus* (Supplementary Figure 1). The five most common species in our metagenomics data were *Bifidobacterium longum*, unclassified *Escherichia* species, *Escherichia coli*, *Bifidobacterium breve*, and *Gemella haemolysans* (Supplementary Figure 1).”

From the perspective of review and reproducible analysis, the results provided for such large study are quite at high-level. Many tools (DADA2-phyloseq, metaphlan) described in the Method section provide rich outputs which are informative and can be provided as supplementary material to help reviewers as well as other researchers. For instance, a PCoA plot could potentially reveal important patterns.

We agree that the tools used provide rich data from our samples. We were unable to utilize many of these tools, such as beta diversity to provide PCoA plots, due to the longitudinal nature of our outcome data and the lack of statistical methods for such data.

Other minor comments:

L60: RTI should give full names spell

We have corrected this and defined RTI in our main text in line 70.

Reviewers' comments:

Reviewer #1 (Remarks to the Author):

I cannot spend more hours trying to improve the writing of this manuscript. See below for the many corrections needed just in the Abstract and Introduction.

Abstract

Line 10. It is unclear that the species are gut microbiome. Suggest replace gut with stool.

Line 11 and 13. The associations are positive. Suggest add the word positive.

Line 12 and 16. Suggest add the words "all-cause" before wheezing and diarrhea. Also, in results risk of wheezing was for subjects "for which a medication was prescribed (RR = 2.00, 95% CI: 78 1.16-3.45) and 86% increase in diarrhea was for subjects requiring a doctor visit." These are critical descriptors to understand the data. Also, in results, authors wrote: "Each doubling in alpha diversity was associated with a 39% increase in having an additional infection or symptom of infection" Whereas in the introduction the authors wrote: "increased risk of [respiratory] infection requiring prescription medicines or symptoms of infection involving a visit to a health care provider..." Which is correct?

Line 16. The study focused on respiratory infections. Suggest add the word respiratory.

Lines 10-12. "Higher infant gut microbiota alpha diversity was associated with an increased risk of later infection or respiratory symptoms, specifically upper respiratory tract infections and among vaginally delivered infants with wheezing and diarrhea." This sentence lacks clarity. Even after revising to replace gut with stool, insert "positive" to describe the association, insert respiratory before infection and all-cause before wheezing and diarrhea, the sentence doesn't make sense.

Introduction

In the introductory paragraph the authors cite multiple papers regarding gut microbiome but this manuscript describes results of stool microbiome. Indeed, later in Suppl Fig 1. (The Figure should be relabeled stool, not gut microbiome.) I am struck by the number of respiratory bacteria in both Figures but particularly in Supple Fig 1B. The authors have failed to understand that their work on stool microbiota identified predominantly respiratory bacteria.

Line 35. Suggest replace gut with stool in describing this study in keeping with the Title and the Results.

Line 36. The study is about respiratory infections and symptoms. Suggest insert the word respiratory.

Line 36. The study is about all-cause wheezing and diarrhea. Suggest insert the words all-cause.

Line 38. The study is not about the general U.S. population. Suggest revising to New Hampshire.

Line 39. The study is not about gut microbiome. Suggest replace with stool.

Line 40 and 41. The study is about respiratory infections. Suggest insert respiratory.

Line 40- 41. "...increased risk of [respiratory] infection requiring prescription medicines or symptoms of infection involving a visit to a health care provider..." The sentence implies analysis identified significant risk associated with respiratory] infection requiring prescription medicines. Where is that analysis in the paper to allow the reader to see distinction between questionnaires reported respiratory infection vs. respiratory infection requiring a prescription medicine? "or symptoms of infection involving a visit to a health care provider" Where is the analysis in the paper to allow the reader to see distinction between questionnaires reporting symptoms vs. symptoms involving a visit to a health care provider?

Line 41. The associations are positive. Suggest insert positive.

Line 41-42. Which associations do the authors mean to say are involved with delivery mode? Is it respiratory] infection requiring prescription medicines? Or, "symptoms of infection involving a visit to a health care provider"? Or is it both?

Line 44-45. "as being related to an increased risk of subsequent respiratory infections in infants' first year of life." The finding reported related to respiratory infections and respiratory symptoms? Or was it only respiratory infections requiring prescription medicine? Or was it respiratory symptoms involving a visit to a health care provider?

Reviewer #2 (Remarks to the Author):

The authors have addressed all my comments. Additionally, breastfeeding is another important effect modifier of gut microbiome and respiratory infection in children. I would like the authors to add a short discussion about this recently published paper [https://www.jacionline.org/article/S0091-6749\(22\)00292-5/fulltext](https://www.jacionline.org/article/S0091-6749(22)00292-5/fulltext) in the minor revision process.

Reviewer #3 (Remarks to the Author):

The authors have made sufficient updates and improvements to address my comments in the revised manuscript. I don't have additional comments

Response to Reviewers

We thank the reviewers for their feedback. Below, we provide a point-by-point response.

Reviewers' comments:

Reviewer #1 (Remarks to the Author):

I cannot spend more hours trying to improve the writing of this manuscript. See below for the many corrections needed just in the Abstract and Introduction.

We have gone through the manuscript and the reviewer's suggestions. We also sent the manuscript to a professional editing service. We hope that it is now communicated clearly.

Abstract

Line 10. It is unclear that the species are gut microbiome. Suggest replace gut with stool.

We used the terminology in widely cited microbiome studies¹⁻³. However, we would be willing to change it if you would like us to do so.

1. Eckburg, P. B. *et al.* Diversity of the Human Intestinal Microbial Flora. *Science* **308**, 1635–1638 (2005).
2. Gill, S. R. *et al.* Metagenomic Analysis of the Human Distal Gut Microbiome. *Science* **312**, 1355–1359 (2006).
3. Yatsunenko, T. *et al.* Human gut microbiome viewed across age and geography. *Nature* **486**, 222–227 (2012).

Line 11 and 13. The associations are positive. Suggest add the word positive.

We added the word “Positive” to the sentence:

“Positive associations were specifically observed with *Veillonella* species among all deliveries and *Haemophilus influenzae* among cesarean-delivered infants.”

Line 12 and 16. Suggest add the words “all-cause” before wheezing and diarrhea. Also, in results risk of wheezing was for subjects “for which a medication was prescribed (RR = 2.00, 95% CI: 78 1.16-3.45) and 86% increase in diarrhea was for subjects requiring a doctor visit.”

We added the words “all-cause” before wheezing and diarrhea in the abstract.

These are critical descriptors to understand the data. Also, in results, authors wrote: “Each doubling in alpha diversity was associated with a 39% increase in having an additional infection or symptom of infection” Whereas in the introduction the authors wrote:” increased risk of [respiratory] infection requiring prescription medicines or symptoms of infection involving a visit to a health care provider...” Which is correct?

The latter is correct for the purpose of general interpretation, but our results from statistical analysis can be interpreted as both risk of respiratory infection or symptom of infection and an additional respiratory infection or symptom of infection. This is due to our Poisson link function in our GEE that allows us to obtain relative risk. We changed the text of this sentence to make our findings clearer:

“Higher infant gut microbiota alpha diversity was associated with an increased risk of infections or respiratory symptoms treated with a prescription medicine, and specifically upper respiratory tract infections. Among vaginally delivered infants, a higher alpha diversity was associated with increased risk of all-cause wheezing treated with a prescription medicine and diarrhea involving a visit to a health care provider.”

Line 16. The study focused on respiratory infections. Suggest add the word respiratory.

We added the word “respiratory” before infection in this sentence.

Lines 10-12. “Higher infant gut microbiota alpha diversity was associated with an increased risk of later infection or respiratory symptoms, specifically upper respiratory tract infections and among vaginally delivered infants with wheezing and diarrhea.” This sentence lacks clarity. Even after revising to replace gut with stool, insert “positive” to describe the association, insert respiratory before infection and all-cause before wheezing and diarrhea, the sentence doesn’t make sense.

Please see response above.

Introduction

In the introductory paragraph the authors cite multiple papers regarding gut microbiome but this manuscript describes results of stool microbiome. Indeed, later in Suppl Fig 1. (The Figure should be relabeled stool, not gut microbiome.) I am struck by the number of respiratory bacteria in both Figures but particularly in Supple Fig 1B. The authors have failed to understand that their work on stool microbiota identified predominantly respiratory bacteria.
Line 35. Suggest replace gut with stool in describing this study in keeping with the Title and the Results.

We address the issue of bacteria commonly found in respiratory bacteria in the second paragraph of our discussion section:

“Our analyses found associations with many bacterial species that are commonly found in oral flora, although these bacteria have also been detected in the gut. An early driver of the gut microbiome is diet. One prospective study found that exclusive breastfeeding was inversely related to lower respiratory tract infections among infants and asthma and allergic rhinitis among children four years of age²⁶. The same study highlighted the potential mediating effect of the gut microbiome on the relationship between exclusive breastfeeding and outcomes. Additionally, infants born operatively may have to acquire such species through breast milk²⁷.”

Line 36. The study is about respiratory infections and symptoms. Suggest insert the word respiratory.

We added the word “respiratory” before infections and symptoms.

Line 36. The study is about all-cause wheezing and diarrhea. Suggest insert the words all-cause.

We added the words “all-cause” before wheezing and diarrhea. We explain that the rest of the paper describes all-cause wheezing and diarrhea outcomes in the introduction:

“Wheeze and diarrhea outcomes for this study included those of any cause.”

We also state in our limitations section of the discussion that we do not know the cause of wheeze and diarrhea outcomes:

“Additionally, we could not differentiate between the various causes of wheezing and diarrhea in our dataset.”

Line 38. The study is not about the general U.S. population. Suggest revising to New Hampshire.

We replaced the phrase “of the US” with “in New Hampshire”.

Line 39. The study is not about gut microbiome. Suggest replace with stool.

Please see response above.

Line 40 and 41. The study is about respiratory infections. Suggest insert respiratory.

We added the word “respiratory” before infections and symptoms.

Line 40- 41. ” ...increased risk of [respiratory] infection requiring prescription medicines or symptoms of infection involving a visit to a health care provider...” The sentence implies analysis identified significant risk associated with [respiratory] infection requiring prescription medicines. Where is that analysis in the paper to allow the reader to see distinction between questionnaires reported respiratory infection vs. respiratory infection requiring a prescription medicine? “ or symptoms of infection involving a visit to a health care provider” Where is the analysis in the paper to allow the reader to see distinction between questionnaires reporting symptoms vs. symptoms involving a visit to a health care provider?

We thank the reviewer for pointing out this error in our writing. All respiratory infections and symptoms of infections are those that required prescription medicines. We reworded the sentence to make our findings clearer:

“Based on amplicon sequence variant (ASV) data generated from 16S rRNA sequencing, higher alpha diversity at six weeks of age was associated with having an additional respiratory infection or symptom of respiratory infection requiring a prescription medicine, with associations varying by delivery mode.”

Line 41. The associations are positive. Suggest insert positive.

We reworded the sentence to make our findings clearer. The words “higher” and “having an additional” now imply a positive association.

Line 41-42. Which associations do the authors mean to say are involved with delivery mode? Is it [respiratory] infection requiring prescription medicines? Or, “symptoms of infection involving a visit to a health care provider”? Or is it both?

Respiratory infections and symptoms of infection are combined into one variable as described in our results section. Therefore, the findings reported by delivery mode relate to this combined variable. As mentioned above, this outcome includes respiratory infections and symptoms of infection requiring a prescription medicine. We reworded the sentence to make our findings clearer:

“Based on amplicon sequence variant (ASV) data generated from 16S rRNA sequencing, higher alpha diversity at six weeks of age was associated with having an additional respiratory infection or symptom of respiratory infection requiring a prescription medicine, with associations varying by delivery mode.”

Line 44-45. “as being related to an increased risk of subsequent respiratory infections in infants’ first year of life.” The finding reported related to respiratory infections and respiratory symptoms? Or was it only respiratory infections requiring prescription medicine? Or was it respiratory symptoms involving a visit to a health care provider?

As mentioned, the outcomes include those requiring a prescription medicine. We reworded the sentence to make our findings clearer:

“... being related to an additional subsequent respiratory infection or symptom of respiratory infection requiring a prescription medicine during an infant’s first year of life.”

Reviewer #2 (Remarks to the Author):

The authors have addressed all my comments. Additionally, breastfeeding is another important effect modifier of gut microbiome and respiratory infection in children. I would like the authors to add a short discussion about this recently published paper [https://www.jacionline.org/article/S0091-6749\(22\)00292-5/fulltext](https://www.jacionline.org/article/S0091-6749(22)00292-5/fulltext) in the minor revision process.

We thank the reviewer for bringing this recently published paper to our attention. We have addressed their findings in the second paragraph of our discussion:

“One prospective study found that exclusive breastfeeding was inversely related to lower respiratory tract infections among infants and asthma and allergic rhinitis among children four

years of age. The same study highlighted the potential mediating effect of the gut microbiome on the relationship between exclusive breastfeeding and outcomes.”

Reviewer #3 (Remarks to the Author):

The authors have made sufficient updates and improvements to address my comments in the revised manuscript. I don't have additional comments

An additional, independent reviewer was asked to comment on Reviewer #1's report and the author's rebuttal. This reviewer gave advice about appropriate terminology confidentially to the editor and the author revised the manuscript accordingly.